# Organic matter loading by hippopotami causes subsidy overload resulting in downstream hypoxia and fish kills

Christopher L. Dutton[1], Amanda L. Subalusky[2], Stephen K. Hamilton[3,4], Emma J. Rosi[2] & David M. Post[1]

Organic matter and nutrient loading into aquatic ecosystems affects ecosystem structure and function and can result in eutrophication and hypoxia. Hypoxia is often attributed to anthropogenic pollution and is not common in unpolluted rivers. Here we show that organic matter loading from hippopotami causes the repeated occurrence of hypoxia in the Mara River, East Africa. We documented 49 high flow events over 3 years that caused dissolved oxygen decreases, including 13 events resulting in hypoxia, and 9 fish kills over 5 years. Evidence from experiments and modeling demonstrates a strong mechanistic link between the flushing of hippo pools and decreased dissolved oxygen in the river. This phenomenon may have been more widespread throughout Africa before hippopotamus populations were severely reduced. Frequent hypoxia may be a natural part of tropical river ecosystem function, particularly in rivers impacted by large wildlife.

[1] Department of Ecology and Evolutionary Biology, Yale University, 165 Prospect Street, New Haven, CT 06511, USA. [2] Cary Institute of Ecosystem Studies, Box AB, Millbrook, NY 12545, USA. [3] W.K. Kellogg Biological Station, Michigan State University, 3700 E. Gull Lake Drive, Hickory Corners, MI 49060, USA. [4] Department of Integrative Biology, Michigan State University, 288 Farm Lane, East Lansing, MI 48824, USA. Correspondence and requests for materials should be addressed to C.L.D. (email: cldutton@gmail.com)

Aquatic ecosystems often receive substantial loading of organic matter and nutrients from natural sources in the watershed as well as from anthropogenic discharges[1–3]. Low levels of loading can stimulate productivity and increase diversity[4]. Higher levels of loading can lead to eutrophication, hypoxia, potential loss of diversity, and altered ecosystem functioning[5–9]. Loading of organic matter and nutrients above a critical threshold results in an overload that switches the system from an aerobic to an anaerobic state[10]. Hypoxia in rivers is uncommon due to the high rates of reaeration in flowing waters, and it is typically associated with high anthropogenic nutrient loading when it does occur[7–9]. However, natural hypoxic events have been documented in some tropical rivers during floodplain inundation after seasonal drying[11–13], and fishes in ecosystems that regularly become hypoxic display a variety of adaptations to endure hypoxic stress[14–16]. Although rare, hypoxia in rivers that do not experience hypoxia regularly can be catastrophic for river biota, often leading to widespread fish kills or other alterations in fish community composition and behavior[17]. Frequent, reoccurring hypoxic events are seldom if ever documented in non-floodplain river ecosystems but may be possible under conditions of very high organic matter loading.

Numerous wildlife species transport substantial amounts of nutrients and organic matter from terrestrial into aquatic ecosystems[18–23]. These resource subsidies can have strong effects on recipient ecosystem function[2,24,25]. The hippopotamus (Hippopotamus amphibius), which has long been recognized as an ecosystem engineer through its grazing and wallowing activities[26–29], transports massive amounts of organic matter and nutrients from terrestrial grazing lands into aquatic ecosystems through egestion and excretion[20]. In East Africa, there are an estimated 70,000 hippopotami, potentially loading 52,800 metric tons year$^{-1}$ of organic matter directly into aquatic ecosystems[20,30]. Laboratory and field studies suggest that these inputs may strongly influence aquatic biogeochemistry and food webs[31–34].

Hippopotami are dependent on water bodies for wallowing during the day to thermoregulate and to protect their sensitive hides from desiccation and ultraviolet exposure[35]. As flows are reduced in the dry season, hippopotami congregate in high densities within the remaining aquatic habitat, hereafter called hippo pools[36]. Hippo pools thus become "hot spots" of biogeochemical cycling, fueled by organic matter and nutrient loading from hippopotamus egestion and excretion[37]. Hippopotamus activity in pools may stir and thus aerate and mix the water column[38,39]. However, without enough aeration, chemical stratification and bottom water anoxia can develop through the decomposition of organic matter and accompanying biochemical oxygen demand (BOD).

The Mara River of East Africa flows through the Maasai Mara National Reserve of Kenya and the Serengeti National Park of Tanzania. There are over 4000 hippopotami in the Kenyan portion of the Mara, distributed across an estimated 171 hippo pools along both tributaries and the main channel (Supplementary Fig. 1)[40]. The hippopotami of the Mara load over 8500 kg of organic matter into the aquatic ecosystem each day[20,40]. The river channel is deeply incised, which is fairly typical geomorphology for rivers in this region[41], so during elevated discharge the river remains within its channel rather than extending onto floodplains[42]. Sufficiently large increases in discharge (>2× above calculated baseflow conditions, defined hereafter as a flushing flow) often result in dissolved oxygen (DO) decreases, sometimes to hypoxic levels, in the river channel.

Here we present experiments and modeling that link hypoxic events in the Mara River system to the flushing of hippo pools. We hypothesized that episodic flushing flows disrupt stratification in hippo pools, flushing out the anoxic bottom water and organic matter and carrying a hypoxic pulse of water through the river downstream of the pools. We documented these hypoxic events by in situ measurements with high-resolution automated sensors and samplers. We used a remote-controlled boat to measure the vertical stratification of hippo pools. We then used microcosms, experimental stream arrays, modeling, and a whole ecosystem manipulation to investigate the mechanistic links between the flushing of hippo pools and the occurrence of hypoxic events in downstream river reaches.

Hypoxic events occur frequently in the Mara River, and their severity is greatest when low flow periods are followed by flow pulses that flush hippo pools, particularly after long intervals since the last flushing flow. Entrainment of anoxic bottom waters from hippo pools during high flows causes episodic oxygen depletion in downstream waters through mixing and high rates of BOD. Both modeling and whole-ecosystem manipulation support the flushing of hippo pools as a primary driver of these river-wide hypoxic events.

## Results

**Hypoxic events in the Mara river.** In 49 out of the 55 flushing flows we documented over 3 years, DO concentrations at the New Mara Bridge (NMB), downstream of all hippo pools in our study region, decreased by 0.04 to 5.5 mg L$^{-1}$ over a range of discharges (Fig. 1a, b; Supplementary Fig. 1). In the 49 flushing flows in which DO declined, discharge increased an average of 32 m$^3$ s$^{-1}$ (range, 4–180 m$^3$ s$^{-1}$), and it took an average of 4 h (range, 45 min to 16 h) from the onset of the flushing flows for the peak discharge to occur. Thirteen of the flushing flows resulted in DO concentrations lower than 2 mg L$^{-1}$ (defined here as hypoxic from the standpoint of stress to fishes), and these low-oxygen conditions persisted for several hours. We documented fish kills during four of the flushing flows (Fig. 1c and Supplementary Fig. 2), and five additional fish kills were reported prior to the onset of water quality monitoring in 2013 (Supplementary Table 1). It is possible that there were more fish kills than we were able to document because the Mara River flows through a remote area, and we have observed that scavengers rapidly consume the fish carcasses associated with fish kills.

We used in situ data to investigate whether the degree of hypoxia resulting from a flushing flow was affected by how much time the pools had to become anoxic since the last flushing (using time since previous flushing and initial DO), how fast the pools were flushed (time to peak discharge), and the influence of discharge on entrainment and dilution of the anoxic hippo pool water (HPW) (initial and peak discharge). We found significant positive effects of peak discharge (multiple linear regression; p-value < 0.05, estimate = 5.41, t-value = 4.68) and time since a previous flushing flow (p-value < 0.05, estimate = 2.52, t-value = 2.80), a significant negative effect of discharge immediately preceding the flushing flow (p-value < 0.05, estimate = −2.03, t-value = −5.63), and a significant interaction between peak discharge and time since a previous flushing flow (p-value < 0.05, estimate = −0.72846, t-value = −2.846). We found no significant effect of the time to peak discharge (p-value = 0.90, estimate = 0.04, t-value = 0.13) or initial DO (p-value = 0.14, estimate = −0.46, t-value = −1.51). The significant interaction term indicates that the impact of peak discharge is reduced when there is a shorter time period since the previous flushing flow. These results indicate that a lower antecedent baseflow coupled with a larger increase in discharge after a longer period of time since the last flushing flow will result in a more severe decrease in DO (adjusted R$^2$ = 0.64), and hence they support our hypothesis that flushing of HPWs explains the downstream decreases in DO.

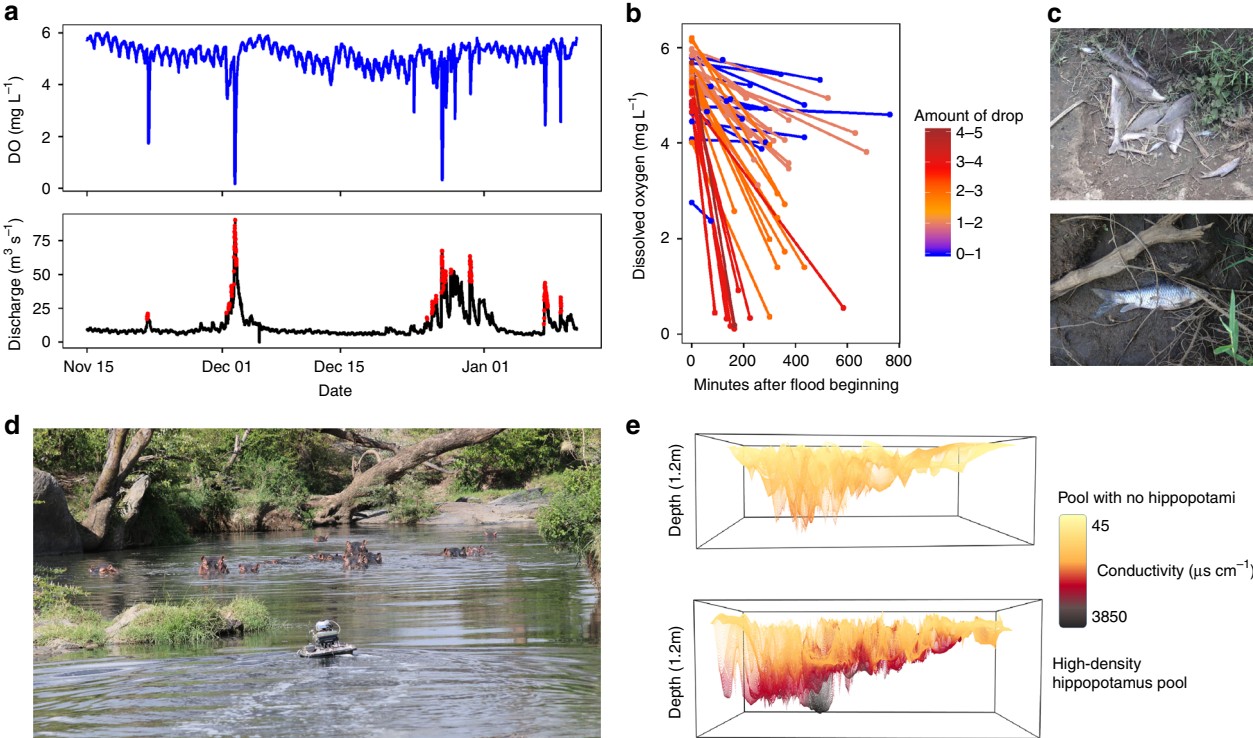

**Fig. 1** Observations of flushing flows, hypoxic events, fish kills, and hippo pools in the Mara River. **a** Dissolved oxygen (blue) and discharge (black) for a 3-month subset of data from November 2014 through January 2015. Red indicates discharge >2 times the calculated baseflow. **b** Dissolved oxygen at the beginning and lowest value for each flushing flow from the 3-year record. **c** Dead fish from a fish kill observed on December 10, 2013 (image credit: Christopher Dutton). **d** Robotic boat surveying a hippo pool (image credit: Amanda Subalusky). **e** 3-D interpolation of the conductivity within pools lacking or containing high densities of hippopotami

**Temporal patterns of flushing flows**. To understand the biogeochemical dynamics during flushing flows, we collected high-frequency river water samples during three flushing flows using automated samplers. We measured DO and water chemistry and estimated the fraction of suspended material comprised of hippopotamus feces using a sediment fingerprinting approach[43]. For all three flushing flows, the lowest DO concentration coincided with the maximum concentration of hippopotamus feces within the suspended material during that flushing flow (Supplementary Figs. 3−5). Total suspended solids, total nitrogen, total phosphorus, and conductivity increased during the flushing flows, and the highest concentrations coincided with both the lowest DO concentrations and the highest concentrations of suspended material derived from hippopotamus feces (Supplementary Note 1).

**Oxygen and chemical profiles of hippo pools**. Our surveys with a remote-controlled boat (Fig. 1d) confirmed that hippo pools can develop chemical stratification in spite of their shallow depths (Fig. 1e and Supplementary Table 2). In oxygen-depleted waters in the hippo pools, we observed accumulation of the products of microbial metabolism, including ammonium, hydrogen sulfide, methane, and carbon dioxide (Supplementary Table 3). Ammonium and sulfide are potentially toxic and thus may contribute to fish mortality during flushing, and the oxidation of those substances as well as methane may contribute to BOD upon reaeration of the water. These concentrations were particularly high in pools along tributaries with low discharge and high numbers of hippopotami (Supplementary Table 3).

**Oxygen depletion assays using microcosms**. The effects of hippo pools on downstream water quality during flushing flows could be due to the mixing and entrainment of the oxygen-depleted water at the bottom of the hippo pools into the river water and/or to the export of labile organic matter and reduced substances into oxic waters that stimulates high rates of oxygen consumption. To ascertain the contribution of these two processes, we conducted a microcosm experiment in bottles by adding unfiltered water from the bottom of HPW, which was anoxic and high in dissolved organic carbon and nutrients, to local groundwater. We added 20% v/v of HPW, which is a conservative estimate of potential contribution of HPW to the river during hypoxic events based on modeling (see Methods). In a second experiment, we added only hippopotamus feces to local groundwater. We left no headspace in the bottles, and we measured the change in DO concentrations over 27 h. The addition of HPW to local groundwater resulted in rapid reduction in DO ($1\,\mathrm{mg\,L^{-1}\,h^{-1}}$) that exceeded the effect due to dilution alone, indicating additional DO consumption. In contrast, the addition of hippopotamus feces to local groundwater resulted in a more protracted reduction in DO ($0.15\,\mathrm{mg\,L^{-1}\,h^{-1}}$) (Fig. 2a).

**Oxygen depletion assays using experimental streams**. Reaeration in flowing waters likely influences the magnitude and duration of the decline in DO concentrations during a flushing flow, so to better simulate a flowing water system, we conducted a second experiment in recirculating experimental stream channels. In this experiment, we added different amounts of HPW and hippopotamus feces, spanning our estimates of potential loading in the Mara River and its tributaries[20,40], to river water in the experimental streams. The addition of HPW to the experimental streams caused a rapid drop in DO ($2.15\,\mathrm{mg\,L^{-1}\,h^{-1}}$ for 10 L of

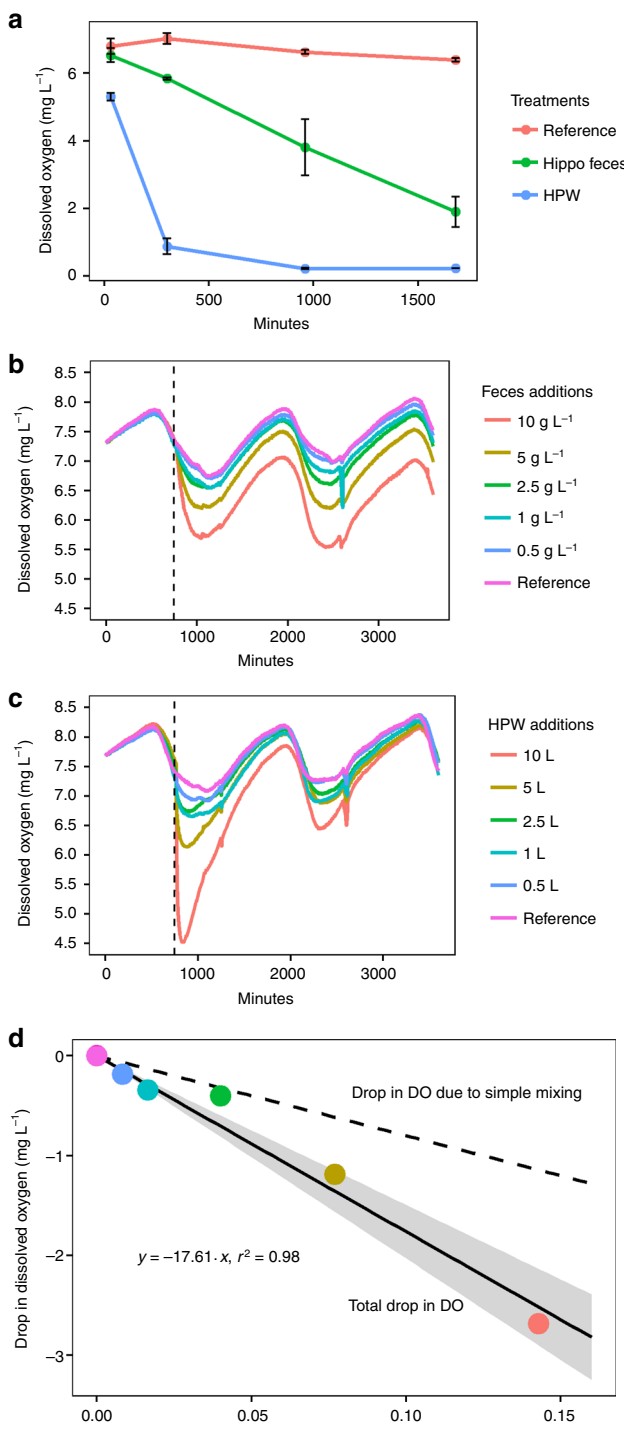

**Fig. 2** Oxygen consumption by river water after addition of either hippopotamus feces or hippo pool water (HPW). **a** Dissolved oxygen for the three treatments in the microcosm experiment (reference, hippopotamus feces, and HPW). Errors bars reflect standard deviation. **b** Dissolved oxygen over 2.5 days for the 6 experimental stream channels with variable amounts of hippopotamus feces added (dashed line indicates time of addition). **c** Dissolved oxygen over 2.5 days for the experimental stream channels with variable amounts of HPW added (dashed line indicates time of addition). **d** The linear relationship between maximum drop in dissolved oxygen and fraction of added HPW is shown as a solid black line with points corresponding to the colors in **c**. 95% confidence intervals shown in gray. The dashed line represents the drop in dissolved oxygen expected for simple mixing of anoxic water with oxic (air-equilibrated) water, without accounting for reaeration, and is thus the minimum possible drop

**Scaling small-scale experiments to the river**. The ameliorating effect of reaeration on DO consumption would be less in a deeper river compared to our experimental streams, and thus a lower fraction of HPW should be sufficient to drive the river to hypoxia. Using a modeling approach to extrapolate the results from experimental streams with 15-cm depth to a river channel with 100-cm depth (similar to the mainstem Mara River), and assuming the same reaeration rates (see Methods and Supplementary Note 2), we estimate that the maximum DO drop in the river would be three-fold greater than in the experimental channels because of the increased depth in the river channel. We thus estimate that, when entrainment of HPW contributes 11% to the Mara River volume during a flushing flow, DO would decline to $2\,\text{mg}\,\text{L}^{-1}$, and when entrainment of HPW reaches 15% of the volume, the river would become anoxic. These estimates are conservative because they assume that the river water upstream of the pools carries DO at atmospheric equilibrium; our data show that DO concentrations are often well below equilibrium at the onset of flushing flows (Fig. 1b), which would exacerbate the effect of HPW on deoxygenation. This estimate of the contribution of HPW necessary to drive the river to hypoxia is plausible in the Mara River. Assuming all hippo pools were flushed and had identical BOD as the HPW used in the experimental stream addition (see Supplementary Table 3), and integrating under an average flushing flow hydrograph, we estimate that water from hippo pools could contribute up to 65% of the total volume of water moving past the NMB site during the average flushing flow, suggesting that there is more than enough HPW to drive the Mara River to hypoxia.

**Experimental test of oxygen depletion at the ecosystem scale**. We conducted a whole-ecosystem manipulation to explicitly test whether flushing a hippo pool during a high flow episode could cause downstream DO depletion. We used a pool that was not frequented by hippopotami but is in the region where they are common. First, we tested the influence of a flushing flow on this "reference pool" by impounding the river water behind a small dam that we constructed upstream of the pool and subsequently releasing the impounded water to flush the pool (reference flushing flow). We then reconstructed the small dam upstream of the pool, and we added 16,000 L of HPW over 2 days to the pool, creating a simulated hippo pool (Fig. 3a, b). We were able to recreate the chemical stratification seen in other hippo pools through the addition of the HPW. We then released the impounded water to flush the simulated hippo pool (treatment flushing flow). During the treatment flushing flow, we labeled the water behind the dam with the tracer dye Rhodamine-WT (RWT) to understand the mixing of the flushing flow water from upstream with the water within the simulated hippo pool.

HPW in 50 L of air-equilibrated river water, 17% v/v of HPW), but DO concentrations subsequently recovered over the 2 days of the experiment as reaeration replenished the DO and oxygen demand diminished (Fig. 2c). The decline in DO concentration was 53% greater than would be predicted based on simple mixing of anoxic HPW with oxic stream water, indicating in situ DO consumption (Fig. 2d). Adding hippopotamus feces to experimental streams resulted in smaller reductions in DO concentrations ($0.06\,\text{mg}\,\text{L}^{-1}\,\text{h}^{-1}$ for $1\,\text{g}\,\text{L}^{-1}$ of hippopotamus feces), but concentrations continued to decrease over 2 days in the highest fecal concentration treatments (5 and $10\,\text{g}\,\text{L}^{-1}$), suggesting more protracted microbial processing of the added organic matter compared to the treatments with only HPW (Fig. 2b).

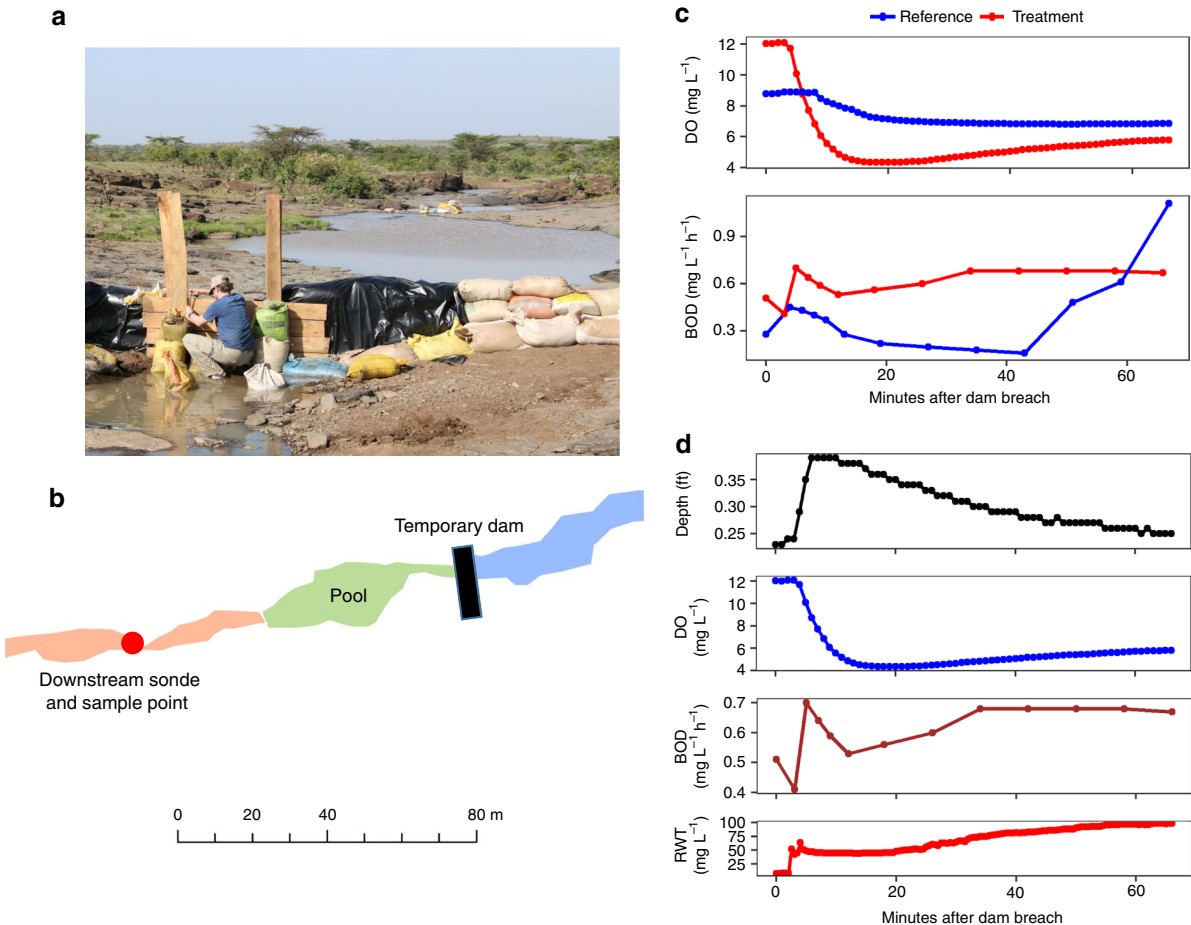

**Fig. 3** Experimental flush of a reference and treatment hippo pool. **a** A temporary dam was built upstream of the pool (image credit: Amanda Subalusky). **b** Map of the pool and sonde/sample location. **c** Dissolved oxygen (DO) and biochemical oxygen demand (BOD) for the whole-ecosystem manipulation immediately after breaching the dam in the reference and treatment flows. **d** Depth, DO, BOD, and Rhodamine tracer (RWT) immediately after the breach of the dam in the treatment flow

DO declined downstream of the pool during both the reference and treatment flows (Fig. 3c, d). The decline in DO was four times greater in the treatment flushing flow than the reference flushing flow. DO in the treatment flushing flow decreased from 12.04 mg L$^{-1}$ to 4.33 mg L$^{-1}$ within 19 min of breaching the dam. Surface DO was higher at the beginning of the treatment flushing flow due to increased algal production caused by the nutrient-rich HPW added to the pool. The BOD of the downstream water decreased by 64% during the reference flushing flow but increased by 33% during the treatment flushing flow (Fig. 3c). In the treatment flushing flow, BOD declined with the initial rise in RWT, then rose quickly and peaked at the moment DO began to decline (Fig. 3d). Based on our mechanistic understanding from the experimental streams, the decline in DO downstream of the pool was likely caused by both the entrainment of oxygen-depleted water from the bottom of the simulated hippo pool as well as the high BOD of that entrained water. This experiment simulates the initial hypoxic event caused by the flushing of a single hippo pool but cannot fully capture the complexity of a flushing event caused by rainfall (e.g., dilution or watershed runoff effects) with the sequential flushing of multiple hippo pools as the flow pulse moves downstream.

## Discussion

In the Mara River basin, very high rates of organic matter and nutrient loading by hippopotami cause a subsidy overload that results in the rapid development of anoxic bottom waters in hippo pools. This degradation of water quality by hippopotami extends downstream when episodic high flows flush the chemically stratified bottom water and hippopotamus feces out of the hippo pools. These flushing flows carry oxygen-depleted water with high BOD downstream through tributaries and into the mainstem Mara River, consuming DO faster than it can be resupplied by reaeration and by upstream inputs of oxygenated waters. The entrainment of bottom waters also carries the byproducts of microbial activity (ammonium, hydrogen sulfide, methane) into downstream reaches. During flushing flows, downstream reaches experience an immediate and rapid decrease in DO attributable to both mixing and oxygen consumption processes, although the effect is eventually diminished as the water moves downstream and DO returns to normal through reaeration. The resuspension and downstream transport of hippopotamus feces with high BOD produces a more protracted reduction in DO that may extend further downstream than the effect of entrainment of oxygen-depleted water alone.

We found that the discharge immediately prior to the flushing flow, peak discharge during the flushing flow, and the time since prior flushing flow predict the magnitude of DO decrease during hypoxic events in the Mara River system. Discharge prior to the flushing flow is likely related to the amount of accumulated organic matter and development of hypoxic waters in pools, as low discharge allows greater accumulation of organic matter and

reduced substances in hippo pools, which can then be mobilized by subsequent high flow pulses. Peak discharge provides information about the ability of high flows to flush the system, as higher discharge flows are likely able to flush a greater proportion of organic material from the bottom of hippo pools than lower discharge flows. Time since prior flushing flow likely reflects the amount of organic material being deposited within the river between flushing flows and suggests these events are largely driven by in-stream processes. Additionally, a significant interaction term between time since last flushing flow and peak discharge indicates that a given flushing flow will cause a greater decline in DO when there has been a longer time since the last flushing flow. We did not observe a clear dilution effect at higher flows, which we would have expected if there was a large impact of overland flow inputs diluting the pulse of hypoxic water.

Organic matter carried into the river by runoff from the landscape as well as increased groundwater inputs could also contribute to the decreased DO observed in the river channel during flushing flows. If landscape runoff carrying organic matter were a major driver of these hypoxic events, we would expect to see a slower drop and more protracted consumption of oxygen in the river. However, the rapid rate of decrease in DO during the rising limb suggest that flushing of the pools is the most important driver of hypoxia. Floodplain inundation is not likely to contribute to these episodes since the Mara River is deeply incised throughout the study area and rarely breaches it banks[42]. While runoff and groundwater may contribute to the hypoxic episodes, modeling indicates that up to 65% of the volume of water during an average flushing flow could be from hippo pools, yet only 11% would be needed to reduce the DO to hypoxia ($<2$ mg L$^{-1}$). However, it is unlikely that all hippo pools would be flushed during the same flushing flow and that their HPW would all have the same BOD as the HPW used in the model and experimental stream addition (see Supplementary Table 3). BOD will likely be higher in hippo pools with greater loading and less flushing and in hippo pools that have not experienced a recent flushing flow. Differences in BOD and its constituents among hippo pools and variation in flushing flows over space and time among the 171 hippo pools in the Mara River and its tributaries likely explain some of the unexplained variability in the response of DO at the NMB during different flushing flow events (Fig. 1b).

We recorded DO at only one site, and the magnitude of drop that we recorded may not have been representative of the total magnitude of drop throughout the river system during the flushing flow. DO could have been continuing to decline during the flow pulse as it moved downstream of our monitoring site, or DO could have already reached the lowest levels upstream of our site and have started to recover. Furthermore, we do not have the ability to determine the origin of the flushing flows. There are multiple tributaries containing hippo pools that could have been flushed during different flow episodes (Upper Mara, Talek, Olare Orok, Ntiakntiak, Moliband, etc.). Rainfall is often highly localized within the study area, differentially affecting individual tributaries and drainages. The spatial timing and propagation of the hypoxic waters associated with these flows warrants further attention.

In most studies of terrestrial subsidies of organic matter and nutrients to aquatic ecosystems, the subsidies have been viewed as enhancing secondary production, including fisheries[25], although there are instances of high organic matter loading in which hypoxia and decreased secondary production can occur[34,44]. In the Mara River system, high concentrations of hippopotami can fundamentally alter the aquatic habitat within hippo pools, which often serve as important dry-season refugia for aquatic biota[45], and flushing flows can export hypoxic water and reduced substances through downstream reaches where fish kills can occur.

Hypoxic events are likely to affect the species composition and abundance of aquatic invertebrates and fishes and may affect the long-term functions of the riverine ecosystem[46]. Sensitive taxa may be lost due to these repeated episodes, whereas more tolerant taxa may be favored, and thus the occurrence of hypoxic events may have lasting effects on community composition. On evolutionary time scales, many species of fishes in tropical floodplain rivers may have evolved in response to frequent hypoxia and thus be adapted to withstand these events[14–16,47]. Increasing habitat heterogeneity due to these hypoxic episodes also could increase biodiversity in the river system to the extent that it results in spatial variation in physicochemical conditions that affect aquatic animal communities. The carcasses from fish kills also may represent a rich resource that benefits a variety of terrestrial and aquatic scavengers[48].

Flushing flow episodes may be important to flush and scour pools in the Mara River and other regions with hippopotami, restoring their habitat value for aquatic biota, while exporting organic matter and nutrients to downstream waters and ultimately leading to higher aquatic productivity in the overall river system. Dams and other engineering works that alter the flow regimes of rivers have been proposed on the Mara River and similar rivers elsewhere in Africa[42,49]. Disruption of the natural flow regime, particularly the attenuation of peak flows that is typical of dammed rivers, may degrade or eliminate the cleansing function of episodic flushing flows and result in greater degradation of the pool habitats.

There remain open questions about the degree to which land use changes in the upper basin may interact with changing nutrient and organic loading from hippopotami to influence long-term river ecosystem dynamics in the Mara River. Although the middle reaches of the Mara River are relatively well protected, studies suggest that land use change and development in the upper Mara River Basin could alter the river's hydrology, leading to more extreme high and low flows[42,50–53]. Additionally, the hippopotamus population in the middle reaches of the Mara River has recently stabilized after a 1500% increase since the first surveys conducted in the 1950s[20,40,54]. The size of pre-colonial populations of hippopotami in the Mara is not known, although globally their current range is a fraction of their historical range due to habitat loss and extirpation of this species by humans[30,35]. Changes in hydrology and hippopotamus populations over time will likely influence the frequency and severity of hypoxic events, although they may interact in complex ways. Furthermore, our research supports the hypothesis that flushing of hippo pools is sufficient to cause hypoxic events in the river, but it does not preclude the additional influence of other anthropogenic factors. Future research in the Mara should continue to investigate the relative contributions of anthropogenic and natural drivers in ecosystem dynamics of this river.

It has been hypothesized that, prior to the extirpation of megafauna in many parts of the world, rivers and other aquatic bodies may have been subjected to higher levels of perturbation by large mammals than currently seen[55,56]. Animals can influence aquatic ecosystems by transporting resources from terrestrial to aquatic ecosystems and altering aquatic habitat structure during feeding and drinking; however, these functions have declined along with populations of many large animals. For example, recent estimates suggest animal nutrient transport has declined to 5–8% of values prior to the late-Quaternary megafaunal extinctions[57]. Consideration of the decline and loss of these functions may fundamentally alter our current conception of the reference state of aquatic ecosystems in the absence of human influence, which may in turn affect restoration targets[56]. Our research suggests that hypoxic episodes due to organic matter and nutrient loading by hippopotami may have also been more common

historically. However, great caution should be used in applying this understanding to current incidences of hypoxia due to anthropogenic loading, which may differ in frequency and periodicity from natural events, occur in ecosystems where organisms have not evolved to withstand this stressor, and include pollution sources that carry additional contaminants with other negative consequences[58,59].

Our findings demonstrate a previously undocumented mechanism by which hippopotami alter the structure and function of aquatic ecosystems by transferring large amounts of organic matter and nutrients from land to pools along rivers, which in turn leads to hypoxic events when the pools are flushed by high flow events. The profound influence of the hippopotamus on terrestrial and aquatic ecosystems was historically more widespread prior to population declines and range contraction[30,35]. Hypoxic events have been sporadically observed in aquatic ecosystems with high livestock influence[7,60–62] and may be an under-reported phenomenon in other aquatic ecosystems with large wildlife populations. Frequent, hypoxic events can thus be a natural part of how river ecosystems functioned before large animal populations were reduced, a possibility that challenges our notions of the reference state of rivers in the absence of human influences[56].

## Methods

**Spatiotemporal occurrence of hypoxic events**. In 2012, we installed a water quality sonde (Manta2, Eureka Environmental, Austin, TX, USA) in a 6-in diameter, 20-ft long galvanized steel pipe that was affixed to the NMB (Latitude, Longitude decimal degrees, −1.54618, 35.01898, Supplementary Fig. 1). NMB is downstream of 171 hippo pools surveyed in 2006[40]. The pipe had multiple openings to allow the free flow of water around the sonde. Data were collected from December 2012 through February 2015 (Supplementary Fig. 2). The sonde took measurements every 15 min of temperature, DO, and water depth. A barometric pressure logger was used to correct the depth measurements (BaroTROLL, In-Situ Inc., Fort Collins, CO, USA). The DO sensor was calibrated monthly.

Water depth measurements were converted to discharge using the equations in Dutton et al.[63]. Baseflow was identified using the Lyne–Hollick recursive digital filter[64,65]. We then identified 55 episodes where the water rose quickly enough to double the calculated base flow (base flow mean index was <0.5, Fig. 1a). We have characterized these 55 episodes as in-channel flushing flows, as they are likely to have enough velocity to mobilize organic material from the benthos of the river. The Mara River is deeply incised and disconnected from the floodplain throughout most of the study area and rarely breaches its banks (Supplementary Fig. 6). During high-flow episodes, the water is constrained within the channel. Out of those 55 flushing flows, 49 caused a decrease of DO in the river ranging from 0.04 to 5.5 mg L$^{-1}$ (Fig. 1b) from the beginning of the flushing flow to the point of lowest DO during the flushing flow. Nine of the flushing flows caused the DO to drop below 1 mg L$^{-1}$ for several hours. Increases in discharge ranged from 4 to 180 m$^3$ s$^{-1}$. Peak discharge ranged from 6 to 197 m$^3$ s$^{-1}$, and 43 out of 49 of the flushing flows had a peak discharge less than 65 m$^3$ s$^{-1}$. The average flushing flow increased three-fold over the calculated baseflow.

We conducted a multiple linear regression to determine factors that may contribute to the magnitude of DO drop using the lm function in R v3.4.2[66]. We used total change in DO as the dependent variable and initial DO, initial discharge, peak discharge, time to peak discharge, and number of hours since the last flushing flow as independent variables. We also included the interaction between time since last flushing flow and peak discharge because the effect of time since last flushing flow may be altered by the magnitude of peak discharge during the flushing flow (the degree of flushing). We did not include total storm size in the model; rather, we included initial discharge, peak discharge and time to peak discharge as variables that are components of total storm size but more explicitly linked to the flushing of hippo pools. All variables were log transformed to ensure normality except for beginning DO, which was already normally distributed. Owing to gaps in data coverage (Supplementary Fig. 2), we were only able to determine the timing since prior flushing flow for 44 of the 49 flushing flows.

**Documentation of fish kills**. We documented nine fish kills in the Mara River around the NMB site from 2009 to 2015 (Supplementary Table 1). These fish kills can be explained by hypoxic episodes we have documented in this study, possibly in combination with concurrent rapid increases in concentrations of hydrogen sulfide and suspended solids (Supplementary Table 3). During one of the fish kills, the Government Chemist of Kenya conducted sampling of fish carcasses and found elevated levels of Karate® Insecticide, suggesting that fish kill may have been caused by pollution from agricultural pesticides. In general, however, the high discharge

and suspended sediments that occur during flushing flows in the Mara would be expected to dilute and bind this pesticide, which should limit its influence.

**Hippo pool stratification**. To investigate the potential for a hippo pool to become stratified, we deployed a remote-controlled boat (Fig. 1d) custom built for this purpose by Platypus LLC (Pittsburgh, PA, USA). The remote-controlled boat was outfitted with a custom-built conductivity/temperature sensor developed by Sodaq (Hilversum, The Netherlands). A pressure sensor was attached to the conductivity sensor to accurately detect the depth at which a reading was taken (MS5803-14BA, Measurement Specialties, Freemont, CA, USA). Both sensors were then integrated into a custom Arduino data logger that we developed at Yale University that also logged GPS coordinates.

The remote-controlled boat was deployed in two pools that did not have hippopotami (Emarti, −1.05788, 35.23111 and Moliband, −1.37388, 35.25827), a pool that had a high density of hippopotami on a tributary of the Mara River (Amani, −1.29539, 35.205) and two pools with low density of hippopotami on the Mara River (Croc, −1.38198, 35.01229 and HPA, −1.3931, 35.02828). Surveyed areas were of a similar size in all five pools.

The conductivity/temperature sensor was successively lowered through the water column at multiple locations within each pool. Average conductivity was calculated for 10 cm at the surface and bottom of all pools (Supplementary Table 2). Data from Emarti and Amani were processed with Surfer and Voxler (Golden Software LLC, Golden, CO, USA) to interpolate between points and generate a three-dimensional representation of pools with and without hippopotami.

The high-density hippo pool exhibited higher conductivity and strong chemical stratification (Supplementary Table 2 and Fig. 1e). In pools with moderate discharge or low densities of hippopotami, bottom water anoxia was not observed. The bottom waters of all the hippo pools (i.e., below the thermocline) had higher conductivity than the surface waters.

**Water sampling during flushing flows**. We utilized an automated water sampler (6712C Compact Portable Sampler, Teledyne ISCO, Lincoln, NE, USA) to collect water samples during three flushing flow episodes at the NMB site. The Mara River at the NMB site is disconnected from the floodplain in a constrained portion of the river with a bedrock substrate. The sampler was placed on high ground several hours in advance of a predicted flushing flow. A polyvinyl chloride (PVC) hose from the sampler was tied off in the water next to a water level switch. The water level switch was installed so that activation of the switch by rising water levels would trigger the pump to begin filling sample bottles housed within the portable sampler. The water level switch was positioned approximately 1 foot above the water level. Ice was added to the sample bottle reservoir to keep the samples cool. The sampler was programmed to immediately begin taking water samples once the water level switch was submerged and then take an additional sample either 15, 30, or 60 min thereafter to catch the rising and falling limb of the flushing flow. The water samples were retrieved the next morning.

Water samples from three flushing flows were captured on October 27–28, 2012; December 10, 2013; and March 14, 2014. A water quality sonde (Manta2, Eureka Environmental, Austin, TX, USA) was installed approximately 0.8 km downstream of the automated sampler for the October 27–28, 2012 flushing flow and logged data every 5 min. The sonde was installed next to the automated sampler for the December 10, 2013 and March 14, 2014 flushing flows and logged data every 15 min. Data from the sonde are presented at the same resolution as data from the water samples captured by the ISCO water sampler.

Water samples were preserved and analyzed for inorganic nutrients (NO$_3^-$, SRP, NH$_4^+$) utilizing the methods described below for the December 10, 2013 and March 14, 2014 flushing flows. For the October 27–28, 2012 flushing flow, NH$_4^+$ was determined fluorometrically in the field[67,68], NO$_3^-$ was determined using cadmium reduction on a flow analyzer in the laboratory[69], and SRP was determined using the molybdate blue method on a flow analyzer in the laboratory[70]. Total nitrogen (TN), total phosphorus (TP), dissolved organic carbon (DOC), and ash-free dry mass (AFDM) were measured as described below. BOD was calculated for the flushing flow on December 10, 2013 using methods described below.

A known volume of water from each sample collected during the flushing flow pulses was filtered through a pre-cleaned and pre-weighed Whatman cellulose nitrate membrane filter (47-mm dia., 0.45 µm pore size, GE Healthcare Life Sciences, Pittsburgh, PA, USA). The filter papers were dried in an oven at 60 °C for 24 h and re-weighed, and the mass of the sediment was measured and was corrected for the weight of the filter paper to calculate total suspended solids (TSS). The sediments on the filter were then analyzed for their elemental composition with microwave-assisted multi-acid total digestion on a Perkin-Elmer ICP-MS at Yale University (New Haven, CT, USA) or with a heat-assisted multi-acid total digestion at Bureau Veritas Acme Labs (Vancouver, British Columbia, Canada). A modified sediment fingerprinting statistical method was then used to apportion the total amount of hippopotamus feces-derived sediment within each sample (Supplementary Tables 9-11 and Supplementary Note 3)[43].

**October 27–28, 2012 flushing flow**. The flushing flow began rising at approximately 21:45 h. The automatic sampler took the first sample at 22:38 h after the water level had risen approximately 15 cm. The discharge rose from 6 m³ s⁻¹ at 21:45 h to 33 m³ s⁻¹ by 01:45 h. At the beginning of the flushing flow, DO was 5 mg L⁻¹. By 04:10 h, the DO had reached a minimum concentration of 1.6 mg L⁻¹. At the point of lowest DO, 32% of the suspended sediments were derived from hippopotamus feces (Supplementary Fig. 3, approximately 90,000 kg h⁻¹). The drop in DO coincided with peaks of TSS, TN, TP, DOC, conductivity, and TDS (Supplementary Table 4). TSS rose to >1000 mg L⁻¹ by 01:00 h and stayed there for over 12 h. Concentrations of $NO_3^-$ and $NH_4^+$ declined during the flow pulse. No fish were reported to have died in this flushing flow.

**December 10, 2013 flushing flow**. The water started rising at 05:30 h and the automatic sampler took its first sample at 06:29 h after the water had risen approximately 30 cm. Discharge increased from 6 m³ s⁻¹ at 05:30 h to a maximum of 36 m³ s⁻¹ at 08:45 h (Supplementary Fig. 4). DO reached its lowest point (0.34 mg L⁻¹) 30 min after the peak of the flushing flow and remained <1 mg L⁻¹ for 210 min. At the DO minimum, over 100,000 kg h⁻¹ of hippopotamus feces-derived sediments were traveling through the system. The drop in DO could partially be explained by the increase in BOD during the peak of the DO crash (Supplementary Fig. 4). The peak drop in DO coincided with the peaks of TSS, $NH_4^+$, TN, TP, DOC, conductivity, and TDS (Supplementary Table 5). TSS remained above 4000 mg L⁻¹ for over 8 h. Game wardens reported that fish began dying around 8:30 h. By 09:00 h, thousands were dead along the banks including *Labeo victorianus*, *Labeobarbus altianalis*, *Barbus* sp., and *Mormyrus kannume*. Most of the dead fish washed downstream with the flow pulse.

**March 14, 2014 flushing flow**. The water started rising at 08:15 h and the automatic sampler took its first sample at 08:57 h after the water had risen approximately 25 cm. Discharge increased from 4 m³ s⁻¹ at 08:15 h to a maximum of 91 m³ s⁻¹ at 18:45 h (Supplementary Fig. 5). DO reached its lowest point (0.4 mg L⁻¹) at 13:15 h. DO remained <1 mg L⁻¹ for 2 h. During the peak of the DO drop, over 300,000 kg h⁻¹ of hippopotamus feces was moving through the system (Supplementary Fig. 5). TSS, TN, and TP all peaked at the same time the DO dropped to its lowest concentration (Supplementary Table 6). DOC, $NO_3^-$, and $NH_4^+$ all declined during the flow pulse. TSS remained above 5000 mg L⁻¹ for at least 3 h (likely much longer), and concentrations were >1000 mg L⁻¹ in every sample we collected (6 h).

**Bottle experiment**. Water was collected from a deep borehole and poured into 36 300 mL glass bottles. HPW and hippopotamus feces were collected as described in Supplementary Note 4. Bottles were then spiked with 60 mL of HPW, 1.5 g wet weight of hippopotamus feces, or left as controls and placed in several inches of water inside of a large black, plastic box. Three bottles from each treatment were sacrificed at each sampling time (30, 300, 960, and 1680 min) to measure DO using an optical DO sensor (ProODO, YSI Incorporated, Yellow Springs, OH, USA). For the first and last sample time (30 and 1680 min), water was collected for analysis ($H_2S$, Fe(II), $NH_4^+$, SRP, $NO_3^-$, $CO_2$, $CH_4$, and $N_2O$) and processed in accordance with the methods described below.

DO decreased in the HPW and hippopotamus feces treatments (Supplementary Table 7). Added HPW caused a rapid drop in DO, whereas added hippopotamus feces caused a slow, linear decline in DO (Fig. 2a). Fe(II) and $N_2O$ were not detectable in any of the treatments. $H_2S$ and $CH_4$ decreased over time in the HPW treatment, presumably due to oxidation. $NH_4^+$ and SRP increased over time in the HPW and hippopotamus feces treatments. $CO_2$ increased over time in the HPW treatment and decreased in the control.

**Experimental streams**. We constructed 12 portable, recirculating experimental streams in the field using flexible PVC plastic. Each stream was filled with river water containing DO at concentrations close to atmospheric equilibrium. The final volume of water in each stream after experimental additions was 60 L. A motor-driven paddle wheel provided recirculating flow in each stream. Optical DO loggers (MiniDOT, PME, Inc., Vista, CA, USA) were placed in each stream and configured to log every 5 min for 2.5 days. HPW and hippopotamus feces were collected as described in Supplementary Note 4. Five streams were spiked with varying weights of fresh hippopotamus feces (30, 60, 150, 300, and 600 g wet weight, ~76% water content) while 5 streams were spiked with varying volumes of anoxic HPW (0.5, 1, 2.5, 5, and 10 L). Two streams were used as controls. The depth of water in the experimental streams was approximately 15 cm, which was comparable to depths between hippo pools on tributaries but much shallower than in the river where typical depths ranged from 0.5 to 1.5 m.

For data analysis, the DO concentrations were adjusted so that each stream started at the same value, applying that same correction to all subsequent measurements. This minor adjustment accounted for variability among sensors. Changes in DO in the control channels were subtracted from changes in the treatments. The minimum oxygen concentrations attained during the initial DO drop, expressed relative to the DO in the reference channel, indicated the magnitude of the drop. A mixing model partitioned the oxygen decline between

simple mixing of anoxic HPW (0 mg L⁻¹) with oxic river water (8 mg L⁻¹) and oxygen consumption processes.

HPW caused a rapid drop in DO within the experimental streams, with the magnitude of the drop linearly related to the quantity of HPW added (Fig. 2c). Mixing of anoxic water with oxic stream water only accounted for approximately 47% of the drop in DO measured at the peak of the drop (Fig. 2d). The majority of the drop in DO was caused by biochemical processes. By 44 h after the additions, DO concentrations in all of the HPW treatment streams had returned to close to atmospheric equilibrium, although this reaeration rate may be slower in the river where river depths are greater.

Hippopotamus feces also caused a drop in DO within the experimental streams (Fig. 2b). The drop was smaller than the addition with HPW, and DO maintained a downward trend for the two larger treatments (5 and 10 g L⁻¹) throughout the experiment.

**Modeling downstream oxygen depletion**. We constructed a model to relate the observed DO sags after HPW additions in the experimental streams to DO concentrations in the deeper Mara River channel under the same mixing conditions (Supplementary Note 2). We used the DO and temperature data collected during the experimental stream experiment and then estimated DO consumption rate for each time interval after accounting for reaeration. We then modeled DO concentrations in the deeper river channel assuming the same temperature, DO consumption rate, and reaeration rate as in the experimental stream experiment. The total area and mean depth of hippo pools along the river provided an estimate of the volume of HPW that would be entrained into the river during flushing flows.

Model results for the two highest fractions of HPW are depicted in Supplementary Fig. 7. We estimate that entrained HPW reaching 17% of the river volume (the same proportion in our highest experimental stream treatment) is sufficient to generate hypoxia (< 2 mg L⁻¹) for 16 h, during which the water would be anoxic for 8 h (Supplementary Fig. 7). Note that the river shows greater departures from atmospheric equilibrium as well as a time lag compared to the experimental stream channels; this is directly due to the greater depth of the water column.

We estimated the area of 14 hippo pools in the Mara River and tributaries using Google Earth (Imagery from Digital Globe and CNES/Airbus), confirming apparent pool boundaries with site visits (Supplementary Table 8). Pool volume was conservatively estimated assuming that the average minimum depth of each pool was 1 m (the amount needed for a hippopotamus to stay partially submerged when laying down). With an average pool volume of 3600 m³ and an estimated 171 pools[40], there is approximately 616,000 m³ of HPW in the Mara River and tributaries upstream of NMB. The average baseflow discharge prior to flushing flows was 15 m³ s⁻¹, and the average increase in discharge was 32 m³ s⁻¹ reaching a peak of 47 m³ s⁻¹ and returning to baseflow within 8.5 h. Given this average flushing flow and assuming that the rate of the flood rise and fall are approximately equal, the total amount of water moving during these 8.5 h is estimated to be 948,600 m³.

$$948,600 \text{ m}^3 = \left(15 \text{ m}^3 \text{ s}^{-1} * 8.5 \text{ h}\right) + \left(\frac{32 \text{ m}^3 \text{ s}^{-1} * 8.5 \text{ h}}{2}\right) \quad (1)$$

Thus the proportion of HPW to total water in the river during an average flushing flow is 65%.

**Whole-ecosystem manipulation**. The Moliband Pool (−1.37388, 35.25827) is located in a private conservancy that allows grazing of cattle by the Maasai tribe. The Moliband Pool has never been occupied by hippopotami (Warden Benson, Naboisho Conservancy, personal communication) likely due to the close proximity of Maasai pastoralists, although the pool appears to be suitable habitat and other pools located nearby are occupied. There are no hippo pools occupied upstream of the Moliband Pool.

A temporary dam was constructed from approximately 6 tons of sandbags 14 m upstream of the Moliband Pool (Fig. 3a, b). A water quality sonde (Manta2, Eureka Environmental, Austin, TX, USA) was placed directly at the pool outlet, approximately 53 m downstream of the temporary dam. Water was allowed to back up behind the dam for 2 days. The dam was then quickly breached, simulating a flushing flow moving through the pool. Water samples were collected at the downstream sonde every few minutes after the flush and analyzed for $H_2S$, Fe(II), $NH_4^+$, SRP, $NO_3^-$, $CO_2$, $CH_4$, $N_2O$, TN, TP, and DOC.

After the flushing flow moved through the system, the dam upstream of the pool was put back in place. A total of 16,000 L of benthic HPW was loaded into the Moliband pool over 2 days. A small dam was put downstream of the treatment pool to prevent the premature release of the added HPW. RWT sensors (Hydrolabs, Hach Company, Loveland, CO, USA) were placed in the deepest part of the pool, one at the pool surface in the center of the pool, and one next to the downstream water quality sonde. In all, 8.1 g of RWT was then added to the upper pool, the water behind the upstream dam, for an approximate concentration of 300 µg L⁻¹ of RWT. The downstream dam was then removed and the upstream dam was quickly breached, and water samples were taken every few minutes in the same manner as the first flush.

DO dropped further in the treatment flush compared to the reference flush (Fig. 3c). BOD, TN, TP, DOC, $NH_4^+$, $CO_2$, $CH_4$, conductivity, and turbidity were higher downstream during the treatment flush.

**Water sample analysis**. We preserved water samples for analysis of inorganic nutrients ($NH_4^+$, SRP, $NO_3^-$) by filtering them through a Supor 0.2 µm polysulfone membrane filter directly into a collection bottle and then freezing them until analysis with a portable flow injection analyzer in the field. Ammonium was analyzed using the gas exchange method[69]. Nitrate+nitrite was analyzed using zinc reduction[71]. Soluble reactive phosphate was analyzed using the molybdate blue method[70].

Dissolved ferrous iron (Fe(II)) was measured on a field spectrophotometer (DR 1900 Portable Spectrophotometer, Hach Company, Loveland, CO, USA) using a colorimetric method modified from Lovley and Phillips[72] and Stookey[73]. After collection, the sample was immediately filtered through a 0.2 µm Supor polysulfone membrane filter into a solution of 50 mM 4-(2-hydroxyethyl)-1-piperazineethanesulfonic acid buffer containing ferrozine (1 g $L^{-1}$).

Hydrogen sulfide ($H_2S$) was also measured colorimetrically on the field spectrophotometer immediately after collection. The sample was filtered through a Whatman GF/F glass fiber filter (GE Healthcare Life Sciences, Pittsburgh, PA, USA) into two scintillation vials and then preserved and analyzed according to the methylene blue method of Golterman[74].

DOC samples were filtered through a pre-combusted, pre-weighed Whatman GF/F glass fiber filter. In all, 100 µL of concentrated sulfuric acid was added to 60 mL of sample for preservation before eventual analysis with a Shimadzu high-temperature, platinum-catalyzed total organic carbon analyzer (Shimadzu, Kyoto, Japan). AFDM of suspended particulate material was determined gravimetrically by drying and combusting the filter.

Samples of unfiltered water (60 mL) were collected for analysis of TN and TP and were preserved by adding 100 µL of concentrated sulfuric acid, followed by eventual alkaline potassium persulfate digestion and analysis on an Astoria-Pacific flow analyzer (Astoria-Pacific, Clackamas, OR, USA).

Water samples for analysis of major ions were filtered through a Whatman GF/F 0.45 µm filter and analyzed on a Dionex ion chromatograph system equipped with membrane suppression and conductivity detection (Dionex, Sunnyvale, CA, USA).

Dissolved gases were extracted from water samples utilizing a static headspace equilibration technique[75]. In short, 115 mL of water was drawn into a 140 mL syringe with a stopcock and 25 mL of ambient air was then drawn into the syringe. The syringe was then gently shaken for 5 min to equilibrate the 25 mL of headspace with the dissolved gases in the sample water. A total of 15 mL of air within the headspace was then injected into evacuated exetainers which were stored in larger vials filled with water. Samples were analyzed by gas chromatography at the Cary Institute for Ecosystem Studies (Millbrook, NY, USA).

BOD was determined for the water samples by incubating a subsample diluted to 300 mL with a dilutant containing phosphate buffer, magnesium sulfate, calcium chloride, and ferric chloride[69]. Samples were incubated for approximately 24 h within glass bottles. DO was measured at the beginning and end of the incubation with an optical DO sensor (ProODO, YSI Incorporated, Yellow Springs, OH, USA).

**Data availability**. All data are available in the supplementary materials or from the Dryad Digital Repository: https://doi.org/10.5061/dryad.nh6hn04.

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

## Acknowledgements

We thank the Government of Kenya and the National Council for Science and Technology for authorizing this research (NCST/RRI/12/1/BS-011/25). The National Museums of Kenya and Laban Njoroge provided assistance with permits and logistics. Support in the field was provided by Brian Heath and the Mara Conservancy. Paul Geemi provided field assistance. Funding was provided by US National Science Foundation grants to D.M.P. and E.J.R. (NSF DEB 1354053 and 1354062); a grant from the National Geographic Society to D.M.P.; grants from the Yale Tropical Resources Institute, the Lindsay Fellowship for Research in Africa, the Yale Institute for Biospheric Studies and the Yale MacMillan Center for International and Area Studies to C.L.D.; and a fellowship from the Robert and Patricia Switzer Foundation to A.L.S. WWF-UK provided funding to maintain one of the water quality sondes.

## Author contributions

C.L.D. and A.L.S. conceived of this study. E.J.R. and D.M.P. provided financial means. All the authors designed the study and collected the data. C.L.D. analyzed the data and prepared the manuscript with input from all authors. All authors contributed to revisions.

## Additional information

**Competing interests:** The authors declare no competing interests.

