## [Peer Review File · Nature Communications]

Reviewers' comments:

Reviewer #1 (Remarks to the Author):

This paper examines the effect of Hippo excretion on oxygen dynamics in the Mara River, located in Kenya and Tanzania. The authors demonstrate that low DO events during flooding result from hippo pools, and these events lead to, at least occasionally, fish kills. The authors argue that this phenomenon is previously undocumented and may change how we think about hypoxic events in natural systems, which are normally associated with anthropogenic impacts.

They conclude:

1. Low DO in the river during flooding is a result of both entrained low D.O. water from hippo pools and increased O₂ consumption from the addition of hippo faeces and labile organic matter from the bottom of the pools;
2. The low DO events, at least sometimes, resulted in fish kills;
3. These dynamics may have been more frequent when hippo populations were higher;
4. Damming of river systems may prevent flooding from "cleansing" organic matter from these pools and create greater degradation of hippo pool habitats.

The authors carried out thorough experiments at multiple scales (bottle, mesocosm, whole-ecosystem) to demonstrate that hypoxia was due to Hippo excretion. These were also done in logistically difficult conditions, and appears to lead to a robust conclusion.

Despite the thoroughness of the scientific approach, the conclusions are not very significant for understanding larger-scale implications of this phenomenon; and in fact, the work leads to a fairly predictable outcome, which one might observe when visiting the hippo pools in person. The broader implications, as the authors present them (particularly conclusions 3 and 4) are largely speculative, and are not the most interesting ones to consider. Furthermore, the authors leave out important perspectives. They imply that hypoxic events are inherently damaging to fish biodiversity, without considering that over evolutionary history, these events may actually increase diversity through creation of habitat variation in time and space, as is the case, for example with hypoxia resulting from tropical river floodplain dynamics. The seasonality of the flooding events may in fact make this phenomenon distinctive from anthropogenic impacts, rather than similar to them. This also has interesting broader scale implications when considering principles of re-wilding, conservation of megafauna, and livestock as alternative vectors of terrestrial C input to aquatic ecosystems. The authors did not discuss these very well, if at all.

In addition to missing an exploration of longer term implications of these natural hypoxic events with respect to mega-fauna conservation, land use change and livestock, the authors also miss an opportunity to put these results in a basin-scale understanding of the processing of terrestrial carbon (from hippo feces) in a river network. This is one of the most interesting implications of this work. The authors seem to imply that they will evaluate this in the introduction with an estimate 3500 kg hippo feces entering the river per day. This leaves me wondering: How much terrestrial carbon does this represent, and how much is respired in the river network, and how much is exported downstream? What does this mean for net ecosystem production of savanna ecosystems?

Many of the ideas about mega-fauna and nutrient transport to aquatic ecosystems, reference states, and climate change have been explored before, at least as conceptual models, e.g. see Brian Moss's paper in *Freshwater Biology* (2015 (60):1964-1976) "Mammals, freshwater reference states, and the mitigation of climate change". This paper lays out an argument that freshwater ecosystems were previously much more nutrient rich (and eutrophic) than we think, precisely due to transfers of terrestrial energy and material by megafauna and suggests that carbon sinks related to this should be restored. It's a bit of a glaring omission that neither this paper, nor its ideas, are included in this

paper. The authors may also wish to see papers related to re-wilding, including Donlan et al. 2005 Re-wilding North America. *Nature* (436:913-914) for other arguments related to restoring (and maintaining) mega-fauna in an evolutionary context.

While the quality of the experimental work and analysis appear to be high, I'm afraid that the conclusions and broader implications of the work do not merit publication in *Nature Communications*, but rather in another high-quality journal in the field (e.g. *Ecology*); further improvement could be made by including a more robust discussion of the broader implications of this interesting phenomenon in DO caused by Hippos.

Reviewer #2 (Remarks to the Author):

In reading and reviewing this paper, I first say how much I appreciate the effort involved in collecting this unique dataset in this severely understudied region. Simple measurements required special work-arounds in this system, and the group were very creative in their approaches. The authors present some compelling data to show that hippos are an important driver of river environmental conditions (primarily oxygen but also carbon and nutrients) in an arid tropical watershed. Their findings are consistent with other work showing strong localized impacts of hippo subsidies, but for the first time they have attempted large scale integration of approaches to examine consequences at larger (and not completely defined) scales. While this represents an interesting and novel study, and with data indicative of a strong influence of these animals on river ecosystems, it is difficult to completely distinguish how much of the effect is directly due to pool flushing relative to other mechanisms that could introduce low DO water and organic matter into the river during storms. This detracts from and weakens the case for publication of the manuscript in its current form. In particular, it is not possible to evaluate how well the experiments are matched (scaled) to real conditions, which is relevant since the authors attempt to use experimental data to interpret field observations. What seems like an insufficiently small volume of hypoxic water and reactive solutes in pools are implicated in large and sustained oxygen depletion during floods involving very large volumes of water. The authors may have the data to more directly support their conclusions, but the current manuscript leaves open too many questions as presented. Two major points are described in more detail below, followed by some other suggestions.

The first issue relates to data from Figure 1b and the supplementary DO time series from three floods that are central to the study. The authors present data from more than 50 floods collected over 3 years. Oxygen concentrations during these floods are a function of three general factors 1) the mixing between runoff from the watershed and the flushing of water from pools, 2) the metabolism (DO consumption) of organic matter from the watershed, and from the pools, and 3) reaeration and shallow ground water inputs. The influence of solutes and low oxygen water flushed from hippo pools are identified in the analyses with support from sediment fingerprinting and experiments. However the influence of dilution from what is likely to be very large runoff volumes would seemingly dominate most floods. Analyses could be done that examine the effect of dilution relative to inputs from pool flushing on dissolved oxygen depressions observed during the duration of floods. Current presentation are limited in their strength in revealing expected impacts of flushing of a very small volume of water from hippo pools relative to the large volume of external flood water moving through the system. I suggest re-analysis of the data with this in mind since the authors appear to have the data necessary to examine multiple variables, and have developed models that could do a much better job with this.

This approach would allow better use of data presented in Fig. 1b. If river DO in this system primarily depends on the amount of runoff from the storm and its oxygen concentration, and the cumulative volume of hypoxic/anoxic pool water flushed, then river DO depressions during floods due to pool

flushing should be determined by the size of runoff event (i.e. discharge) and the hypoxic pool volume. Assuming hippo densities and pool abundances are relatively constant, pool conditions might also be influenced by the time between rain events (with longer intervals allowing more time for organics to accumulate and DO to draw down). If the preceding conditions apply on average in the system, then I would expect to see a relationship between discharge and total integrated deviation of river DO from saturation that could be accounted for by dilution and metabolism. Presumably data could be presented this way, and the arguments would be more convincing if such a model of river DO was presented.

In summary, the analysis would be more convincing if they showed under what conditions that the volume of water in pools had sufficient hypoxic water and BOD to account for the observed sustained DO sags. This would help identify other potential mechanisms, such as flushing of shallow groundwater, or overland flow (which could mobilize feces and other organic matter from near stream areas) that might also contribute to low DO conditions during storms.

A second point is closely related to first. Use of experiments and integration of experimental results with modeling to examine the influence of hippos on river oxygen is unique, difficult, and important-well done. That said, the authors have not made a convincing case that they have scaled hippo pool water and feces additions in their experiments (bottles, small channels and whole pool) appropriately to the dilution that would occur during watershed scale floods. The use of experimental results in the models is absolutely dependent on the assumption is made regarding the amounts of oxygen and carbon from hippo pools relative to experimental flood water volume. If the authors can elaborate on their approach and provide more convincing data to support use of these results in scaling up, it would make them much more helpful in support of their arguments.

Two other areas that deserve some thought in a revision of this manuscript.

1) The statement that the results of this study "...directly challenge our notions of the pristine conditions of rivers in the absence of human influences" is problematic and must be clarified. While this may be true in some cases for tropical rivers, it does not apply to all of them. Importantly, I worry that this statement could easily be taken out of context by parties wishing to circumvent or otherwise avoid restrictions on polluting activities (i.e. mining, agricultural companies). Please revise to avoid this unintended (potential) use of this statement in the abstract and discussion.

2) Statements starting on line 43 through 49 are seemingly contradictory and will be confusing. River hypoxia is stated to be rare, with a few exceptions. A following statement indicated that such events lead to widespread impacts, a statement supported by 6 citations. These statements can easily be revised to provide a clear picture of the low frequency of such events documented in rivers, and the likelihood of large impacts when they do occur.

In addition, information about the location of the dissolved oxygen logger relative to the spatial distribution of upstream pools is a critical piece of information (see above comments) that should be provided in the methods and possibly also in a supplemental figure.

Reviewer #3 (Remarks to the Author):

General comments:

This study describes how hippopotami affect water quality, in particular the dissolved oxygen concentration (DO). The study fits in a recent series of studies into the ecological functions of hippos. However, whereas previous studies addressed the role of nutrient transport or impacts on terrestrial

habitats, the impact of hippo's on DO is entirely novel. I am aware of only one study that has ever examined this, but at a very limited scale (only during 24 hours) and purely descriptive where the conclusion was that water movement by hippo's increased DO, if I remember it correctly (the study is cited as reference 34 in the present paper). Whereas it is well known that loading of aquatic systems with terrestrially derived organic material causes oxygen consumption, which may eventually lead to anoxia. In this respect the present study is one in the category: why has nobody ever thought of this before? Considering the enormous input of organic material by hippo's inevitably there has to be ecosystem impact. Previously this has been described by hippo's fueling the aquatic food web with energy. However, not all impacts of charismatic animals are necessarily beneficial to the rest of the ecosystem, in this case hippo dung causing anoxia leading to fish kills. Although eventually this fuels the scavenger community, hence there are benefits through the circle of life. In this respect the study reminded me of the recent study of Subalusky et al (2017) PNAS on the nutrient supplementation of dying wildebeest during crossing the river to the aquatic ecosystem and the scavenger community. However, the present study on hippo impacts is very different from that study and I personally would estimate that the hippo impacts would be much larger considering their territorial presence along many African rivers, particularly in historic times. I am very positive about this study, which I think is ground breaking in:

- The topic addressed: who would ever think of a megaherbivore causing fish kills, an exemplary case of indirect interactions, that goes way beyond the hippo example
- The extremely creative use of methods: from descriptive (measuring actual flooding events), to experimental (in microcosms to show BOD and flume tanks to show BOD under flooding regimes both with varying amounts of hippo dung and water from hippo pools) to modeling to the field experiment. I am an experimental ecologist not afraid of large scale experiments, but have to acknowledge defeat in this hippo artificial pond break-through flooding experiment, it is very large-scale and simply brilliant. It demonstrates the point beyond any discussion.
- Figure 1D: really, there is a reason why people don't study hippo's directly, but rather collect their dung or so when they are far away, they are straight dangerous. To come up with this robot boat is so cool and so effective – an example for many more aquatic studies.
- Providing a new baseline in our thinking about pristine conditions of aquatic ecosystems. Most of the aquatic ecosystems worldwide are devoid of the influence of large herbivores, let alone megafauna. This has resulted in a line of thinking that in many places of the world the pristine ecosystem state is rather oligotrophic, and eutrophication is a human-induced phenomenon. In this sense the study fits perfectly in the defaunation discussion, e.g. that the loss of wildlife has caused a new ecosystem state which is far from pristine, since a major component, e.g. the wildlife, is missing from it. In this respect the present study goes far beyond hippo's as this applies to large parts of the world.

Then here I come to a small missing part, I'm missing a reference to the following paper: Moss, B. (2015) Mammals, freshwater reference states, and the mitigation of climate change. *Freshwater Biology* 60: 1964–1976, DOI: 10.1111/fwb.12614. I don't know whether the authors are familiar with it, I think it is highly relevant for the end conclusion, as Brian Moss says in this paper that the reference state of freshwater bodies (as embodied by the Water Framework Directive in Europe) is only valid if you exclude the role that large mammals have played before they were hunted to extinction in most parts of the world, with strong reference to the African situation. I think the current paper is highly complementary to Moss, and providing much more data, a quantitative approach, and experimental proof, since Moss was an opinion paper. Still it would be elegant to refer to it, also to clearly demonstrate that the whole point of the paper is much beyond areas where hippo's occur.

Specific comments:

- Throughout the paper there are hardly any statistics. Often the patterns are clear from the figures, but some statistical back-up would be elegant, particularly since the quantitative aspect of the paper is one of its strongest assets. Low-hanging fruit for instance the correlation between DO and discharge from Fig. 1A, test of data in Fig. 2A (and add error bars reflect xxx in caption).

- Suppl. Table 2: would be helpful to add coordinates of pools
- Suppl. Table 3: elegant to add K-value and df
- Suppl. Table 6: give location
- Reference list: remove capital letters
- Reference 35 and 49 seem the same

Responses to reviewers are bulleted inline below. Relevant changes in the manuscript are highlighted.

Reviewers' comments:

Reviewer #1 (Remarks to the Author):

This paper examines the effect of Hippo excretion on oxygen dynamics in the Mara River, located in Kenya and Tanzania. The authors demonstrate that low DO events during flooding result from hippo pools, and these events lead to, at least occasionally, fish kills. The authors argue that this phenomenon is previously undocumented and may change how we think about hypoxic events in natural systems, which are normally associated with anthropogenic impacts. They conclude:

1. Low DO in the river during flooding is a result of both entrained low D.O. water from hippo pools and increased O₂ consumption from the addition of hippo faeces and labile organic matter from the bottom of the pools;
2. The low DO events, at least sometimes, resulted in fish kills;
3. These dynamics may have been more frequent when hippo populations were higher;
4. Damming of river systems may prevent flooding from “cleansing” organic matter from these pools and create greater degradation of hippo pool habitats.

The authors carried out thorough experiments at multiple scales (bottle, mesocosm, whole-ecosystem) to demonstrate that hypoxia was due to Hippo excretion. These were also done in logistically difficult conditions, and appears to lead to a robust conclusion.

- Thank you for taking the time to review our manuscript. We appreciate the feedback you provided.

Despite the thoroughness of the scientific approach, the conclusions are not very significant for understanding larger-scale implications of this phenomenon; and in fact, the work leads to a fairly predictable outcome, which one might observe when visiting the hippo pools in person.

- It is evidently not as obvious as one might think. We would like to point out that our results contradict the only other published observation of hippo effects on water quality that we know of, one that suggested that hippos aerate the water where they wallow, an idea that has often been cited in papers and textbooks.

The broader implications, as the authors present them (particularly conclusions 3 and 4) are largely speculative, and are not the most interesting ones to consider. Furthermore, the authors leave out important perspectives. They imply that hypoxic events are inherently damaging to

fish biodiversity, without considering that over evolutionary history, these events may actually increase diversity through creation of habitat variation in time and space, as is the case, for example with hypoxia resulting from tropical river floodplain dynamics. The seasonality of the flooding events may in fact make this phenomenon distinctive from anthropogenic impacts, rather than similar to them. This also has interesting broader scale implications when considering principles of re-wilding, conservation of megafauna, and livestock as alternative vectors of terrestrial C input to aquatic ecosystems. The authors did not discuss these very well, if at all.

- We have strengthened our discussion to specifically discuss how these events may affect the long-term functions of the riverine ecosystem (Lines 293-305). We have also included text highlighting your point that these events may actually lead to increased diversity due to habitat heterogeneity (Lines 302-304).
- We have expanded our discussion on the broader scale implications of these events concerning the perception of reference states of rivers prior to megafaunal extinctions and widespread reductions of extant large herbivore populations (Lines 317-332).

In addition to missing an exploration of longer term implications of these natural hypoxic events with respect to mega-fauna conservation, land use change and livestock, the authors also miss an opportunity to put these results in a basin-scale understanding of the processing of terrestrial carbon (from hippo feces) in a river network. This is one of the most interesting implications of this work. The authors seem to imply that they will evaluate this in the introduction with an estimate 3500 kg hippo feces entering the river per day. This leaves me wondering: How much terrestrial carbon does this represent, and how much is respired in the river network, and how much is exported downstream? What does this mean for net ecosystem production of savanna ecosystems?

- The development of a full understanding of the processing of terrestrial carbon throughout a river basin network influenced by large wildlife would be a fantastic contribution to the literature. However, it is beyond the scope of this manuscript. The focus of this manuscript is on understanding the processing of this carbon within the river channel and its contribution to frequent dissolved oxygen crashes. These events have never been documented before because of their ephemeral nature (they happen quickly and leave little outward signs) and the overall paucity of high resolution measurement of water quality in rivers dominated by hippopotami. There is some information on the influence of hippos in this river in comparison to upstream sources of carbon and nutrients, and to the translation of these inputs into savanna productivity, in our paper in *Freshwater Biology* (Subalusky et al. 2014), which we have cited in the Introduction.

Many of the ideas about mega-fauna and nutrient transport to aquatic ecosystems, reference states, and climate change have been explored before, at least as conceptual models, e.g. see Brian Moss's paper in *Freshwater Biology* (2015 (60):1964-1976) "Mammals, freshwater reference states, and the mitigation of climate change". This paper lays out an argument that freshwater ecosystems were previously much more nutrient rich (and eutrophic) than we think, precisely due to transfers of terrestrial energy and material by megafauna and suggests that carbon sinks related to this should be restored. It's a bit of a glaring omission that neither this paper, nor its ideas, are included in this paper. The authors may also wish to see papers related to re-wilding, including Donlan et al. 2005 Re-wilding North America. *Nature* (436:913-914) for other arguments related to restoring (and maintaining) mega-fauna in an evolutionary context.

- We are familiar with Brian Moss's paper and its very interesting ideas, as it cites our earlier work investigating hippopotami as vectors of carbon and nutrient loading in the Mara River (Subalusky et al. 2014). We have included additional discussion on the idea of reference states and reference to this paper in Lines 317-332.

While the quality of the experimental work and analysis appear to be high, I'm afraid that the conclusions and broader implications of the work do not merit publication in *Nature Communications*, but rather in another high-quality journal in the field (e.g. *Ecology*); further improvement could be made by including a more robust discussion of the broader implications of this interesting phenomenon in DO caused by Hippos.

- Thank you for the time to review our manuscript. We have expanded our discussion section to discuss the broader implication of our research. We have incorporated your suggestions and we feel that the manuscript is greatly improved.

Reviewer #2 (Remarks to the Author):

In reading and reviewing this paper, I first say how much I appreciate the effort involved in collecting this unique dataset in this severely understudied region. Simple measurements required special work-arounds in this system, and the group were very creative in their approaches. The authors present some compelling data to show that hippos are an important driver of river environmental conditions (primarily oxygen but also carbon and nutrients) in an arid tropical watershed. Their findings are consistent with other work showing strong localized impacts of hippo subsidies, but for the first time they have attempted large scale integration of approaches to examine consequences at larger (and not completely defined) scales. While this represents an interesting and novel study, and with data indicative of a strong influence of these animals on river ecosystems, it is difficult to completely distinguish how much of the effect is directly due to pool flushing relative to other mechanisms that could introduce low DO

water and organic matter into the river during storms. This detracts from and weakens the case for publication of the manuscript in its current form. In particular, it is not possible to evaluate how well the experiments are matched (scaled) to real conditions, which is relevant since the authors attempt to use experimental data to interpret field observations. What seems like an insufficiently small volume of hypoxic water and reactive solutes in pools are implicated in large and sustained oxygen depletion during floods involving very large volumes of water. The authors may have the data to more directly support their conclusions, but the current manuscript leaves open too many questions as presented. Two major points are described in more detail below, followed by some other suggestions.

- Thank you for taking the time to review our manuscript. Below, we provide the specific ways that we have addressed your concerns.

The first issue relates to data from Figure 1b and the supplementary DO time series from three floods that are central to the study. The authors present data from more than 50 floods collected over 3 years. Oxygen concentrations during these floods are a function of three general factors 1) the mixing between runoff from the watershed and the flushing of water from pools, 2) the metabolism (DO consumption) of organic matter from the watershed, and from the pools, and 3) reaeration and shallow ground water inputs. The influence of solutes and low oxygen water flushed from hippo pools are identified in the analyses with support from sediment fingerprinting and experiments. However the influence of dilution from what is likely to be very large runoff volumes would seemingly dominate most floods. Analyses could be done that examine the effect of dilution relative to inputs from pool flushing on dissolved oxygen depressions observed during the duration of floods. Current presentation are limited in their strength in revealing expected impacts of flushing of a very small volume of water from hippo pools relative to the large volume of external flood water moving through the system. I suggest re-analysis of the data with this in mind since the authors appear to have the data necessary to examine multiple variables, and have developed models that could do a much better job with this.

- We have made several changes in the manuscript in an effort to be more clear and to provide some additional information about potential drivers of these hypoxic events.
 - In our original draft, the use of the term “flood” for high flow events may have created a misleading impression of much larger volumes of water, with the river flooding over its banks. We have clarified that high flows and subsequent hypoxic events in the Mara River are not associated with floodplain inundation by including a more detailed description of the Mara River and by changing our terminology from “flood” to “flushing flow”. The Mara River is deeply incised (Supplementary Figure 3) and rarely breaches its banks. None of the events we captured would have overtopped the banks of the river. The word “flood” often

refers to the flooding of a floodplain, and that is not what we are experiencing in the Mara as it is disconnected from its floodplain. To prevent this misconception, we are now using the term “flushing flow” to describe these events, which is an increase in discharge sufficient to flush bottom water and material from the pools into downstream reaches of the river (Lines 78-81, 363-368).

- There is no evidence that shallow groundwater input would make a significant contribution to the discharge during high flow events. If it were important there would be visible seeps and springs along the banks at low water levels.
- We have added a multiple regression to the manuscript that examines the role of several potential predictors of variation in the magnitude of declines in DO during flushing flows, including peak discharge, discharge immediately preceding the flushing flow, time to peak discharge, time since a previous flushing flow and beginning DO. Several factors were significant and there was no clear dilution effect observed at higher flows in which the DO decreased is obscured. More on this is presented below.
- Our estimates of fractional contributions of river flows from upstream mixing with pool waters show that the flushing of the pools together with the oxygen demand of that water could easily explain the river DO decreases, and hence the dilution effect is not as large as one might imagine from our original description of the river hydrology. We have added additional data in the manuscript on the number and spatial location of 171 hippopotamus pools surveyed in 2006 (Supplementary Fig. 1). Additionally, we have added information on the size and volume of water in 14 different hippopotamus pools (Lines 695-710, Supplementary Table 11) to calculate an average hippo pool volume. With this new data, we show that during an average flushing flow, up to 65% of the water that passes the NMB site could be from a hippopotamus pool yet only 15% is needed to drive the river to anoxia (Lines 196-201).
- We have added additional discussion about this in the manuscript (Lines 270-280), copied below:

Organic matter carried into the river by runoff from the landscape as well as increased groundwater inputs could also contribute to the hypoxia observed in the river channel during flushing flows. If landscape runoff carrying organic matter were a major driver of these hypoxic events, we would expect to see a slower drop and more protracted consumption of oxygen in the river. However, the rapid rate of decrease in DO during the rising limb suggest that flushing of the pools is the most important driver of hypoxia. Floodplain inundation is not likely to contribute to these episodes since the Mara River is deeply incised throughout the study area and rarely breaches its banks. While runoff and groundwater may

contribute to the hypoxic episodes, modeling indicates that over 60% of the volume of water during an average hypoxic episode could be from the flushing of hippo pools, yet only 11% would be needed to reduce the DO to 2 mg L⁻¹.

This approach would allow better use of data presented in Fig. 1b. If river DO in this system primarily depends on the amount of runoff from the storm and its oxygen concentration, and the cumulative volume of hypoxic/anoxic pool water flushed, then river DO depressions during floods due to pool flushing should be determined by the size of runoff event (i.e. discharge) and the hypoxic pool volume. Assuming hippo densities and pool abundances are relatively constant, pool conditions might also be influenced by the time between rain events (with longer intervals allowing more time for organics to accumulate and DO to draw down). If the preceding conditions apply on average in the system, then I would expect to see a relationship between discharge and total integrated deviation of river DO from saturation that could be accounted for by dilution and metabolism. Presumably data could be presented this way, and the arguments would be more convincing if such a model of river DO was presented.

- We greatly appreciate your suggestion to add additional analysis of our long-term data. As noted above, we have added a multiple regression to the manuscript to examine the factors that predict the magnitude in observed DO decreases. We used total change in DO as the dependent variable and the following independent variables: beginning DO, time to peak discharge, beginning discharge, number of hours since the last flushing flow and peak discharge. An interaction term was also included in the multiple regression between number of hours since the last flushing flow and peak discharge. The interaction term and all variables except for beginning DO (p-value = 0.14) and time to peak (p-value=0.87) were significant (p-value < 0.05). The multiple regression has an adjusted R² of 0.64. We have added additional text about this in Lines 110-126, 253-268, and 374-384.

In summary, the analysis would be more convincing if they showed under what conditions that the volume of water in pools had sufficient hypoxic water and BOD to account for the observed sustained DO sags. This would help identify other potential mechanisms, such as flushing of shallow groundwater, or overland flow (which could mobilize feces and other organic matter from near stream areas) that might also contribute to low DO conditions during storms.

- We have addressed this in the above comments.

A second point is closely related to first. Use of experiments and integration of experimental results with modeling to examine the influence of hippos on river oxygen is unique, difficult,

and important- well done. That said, the authors have not made a convincing case that they have scaled hippo pool water and feces additions in their experiments (bottles, small channels and whole pool) appropriately to the dilution that would occur during watershed scale floods. The use of experimental results in the models is absolutely dependent on the assumption is made regarding the amounts of oxygen and carbon from hippo pools relative to experimental flood water volume. If the authors can elaborate on their approach and provide more convincing data to support use of these results in scaling up, it would make them much more helpful in support of their arguments.

- With the inclusion of the additional data on the location and number of hippopotamus pools and the approximate volume of an average hippopotamus pool (Lines 695-710, Supplementary Fig. 1 and Supplementary Table 11), we are able to calculate the percentage of hippopotamus pool water moving through our downstream site (NMB) during the average flushing flow. Hippopotamus pools could contribute up to 65% of the water moving past NMB during an average flushing flow (Lines 196-201). Of course, not all hippo pools are the same. High density hippo pools on tributaries likely have much stronger chemical stratification and higher bottom-water biochemical oxygen demand than low density hippo pools in the Mara River. In our bottle experiment and artificial stream experiment, we used conservative amounts of HPW within the range of possibilities encountered during these flushing flows (Lines 156-158, 168-171, 196-201). The bottle experiment used 20% and the artificial streams used approximately 1% to 17% hippopotamus pool water. Thank you for the suggestion and hope that the addition of this information makes it clear how our experiments are appropriately scaled for the Mara River.

Two other areas that deserve some thought in a revision of this manuscript.

1) The statement that the results of this study “...directly challenge our notions of the pristine conditions of rivers in the absence of human influences” is problematic and must be clarified. While this may be true in some cases for tropical rivers, it does not apply to all of them. Importantly, I worry that this statement could easily be taken out of context by parties wishing to circumvent or otherwise avoid restrictions on polluting activities (i.e. mining, agricultural companies). Please revise to avoid this unintended (potential) use of this statement in the abstract and discussion.

- Good point. We have revised this sentence and it now reads, “...directly challenges our notions of the reference state of rivers in the absence of human influences” (Lines 34-35). We also included additional discussion on the idea of reference states (Lines 317-332) as well as a caveat to address your concern, Lines 328-332: “*However, great caution should be used in applying this understanding to current incidence of hypoxia*”

due to anthropogenic loading, which may occur in ecosystems where organisms have not evolved to withstand this stressor, and where pollution sources may carry additional contaminants with other negative consequences^{44,45}.”

2) Statements starting on line 43 through 49 are seemingly contradictory and will be confusing. River hypoxia is stated to be rare, with a few exceptions. A following statement indicated that such events lead to widespread impacts, a statement supported by 6 citations. These statements can easily be revised to provide a clear picture of the low frequency of such events documented in rivers, and the likelihood of large impacts when they do occur.

- We have revised those statements to be more clear (Lines 43-50).

In addition, information about the location of the dissolved oxygen logger relative to the spatial distribution of upstream pools is a critical piece of information (see above comments) that should be provided in the methods and possibly also in a supplemental figure.

- We have clarified in the Results (Lines 96-97) and Methods (Line 353) that the dissolved oxygen logger is downstream of all the hippopotamus pools surveyed in Kanga (2011). We have created a map to explicitly show the location of the dissolved oxygen logger in relation to surveyed hippopotamus pools as well as the hippopotamus pools mentioned in this manuscript (Supplementary Figure 1).

Reviewer #3 (Remarks to the Author):

General comments:

This study describes how hippopotami affect water quality, in particular the dissolved oxygen concentration (DO). The study fits in a recent series of studies into the ecological functions of hippo's. However, whereas previous studies addressed the role of nutrient transport or impacts on terrestrial habitats, the impact of hippo's on DO is entirely novel. I am aware of only one study that has ever examined this, but at a very limited scale (only during 24 hours) and purely descriptive where the conclusion was that water movement by hippo's increased DO, if I remember it correctly (the study is cited as reference 34 in the present paper). Whereas it is well known that loading of aquatic systems with terrestrially derived organic material causes oxygen consumption, which may eventually lead to anoxia. In this respect the present study is one in the category: why has nobody ever thought of this before? Considering the enormous input of organic material by hippo's inevitably there has to be ecosystem impact. Previously this has been described by hippo's fueling the aquatic food web with energy. However, not all impacts of charismatic animals are necessarily beneficial to the rest of the ecosystem, in this

case hippo dung causing anoxia leading to fish kills. Although eventually this fuels the scavenger community, hence there are benefits through the circle of life. In this respect the study reminded me of the recent study of Subalusky et al (2017) PNAS on the nutrient suppletion of dying wildebeest during crossing the river to the aquatic ecosystem and the scavenger community. However, the present study on hippo impacts is very different from that study and I personally would estimate that the hippo impacts would be much larger considering their territorial presence along many African rivers, particularly in historic times. I am very positive about this study, which I think is ground breaking in:

- The topic addressed: who would ever think of a megaherbivore causing fish kills, an exemplary case of indirect interactions, that goes way beyond the hippo example
- The extremely creative use of methods: from descriptive (measuring actual flooding events), to experimental (in microcosms to show BOD and flume tanks to show BOD under flooding regimes both with varying amounts of hippo dung and water from hippo pools) to modeling to the field experiment. I am an experimental ecologist not afraid of large scale experiments, but have to acknowledge defeat in this hippo artificial pond break-through flooding experiment, it is very large-scale and simply brilliant. It demonstrates the point beyond any discussion.
- Figure 1D: really, there is a reason why people don't study hippo's directly, but rather collect their dung or so when they are far away, they are straight dangerous. To come up with this robot boat is so cool and so effective – an example for many more aquatic studies.
- Providing a new baseline in our thinking about pristine conditions of aquatic ecosystems. Most of the aquatic ecosystems worldwide are devoid of the influence of large herbivores, let alone megafauna. This has resulted in a line of thinking that in many places of the world the pristine ecosystem state is rather oligotrophic, and eutrophication is a human-induced phenomenon. In this sense the study fits perfectly in the defaunation discussion, e.g. that the loss of wildlife has caused a new ecosystem state which is far from pristine, since a major component, e.g. the wildlife, is missing from it. In this respect the present study goes far beyond hippo's as this applies to large parts of the world.

- Thank you for taking the time to review our manuscript.

Then here I come to a small missing part, I'm missing a reference to the following paper: Moss, B. (2015) Mammals, freshwater reference states, and the mitigation of climate change. *Freshwater Biology* 60: 1964–1976, DOI: 10.1111/fwb.12614. I don't know whether the authors are familiar with it, I think it is highly relevant for the end conclusion, as Brian Moss says in this paper that the reference state of freshwater bodies (as embodied by the Water Framework Directive in Europe) is only valid if you exclude the role that large mammals have played before they were hunted to extinction in most parts of the world, with strong reference to the African situation. I think the current paper is highly complementary to Moss, and providing much more data, a quantitative approach, and experimental proof, since Moss was an opinion paper. Still it would be elegant to refer to it, also to clearly demonstrate that the whole point of the paper is much beyond areas where hippo's occur.

- We are familiar with Brian Moss's paper and its very interesting ideas, as it cites our earlier work investigating hippopotami loading of carbon and nutrients into the Mara River (Subalusky et al. 2014). We have included additional discussion on the idea of reference states and reference to this paper in Lines 317-332.

Specific comments:

- Throughout the paper there are hardly any statistics. Often the patterns are clear from the figures, but some statistical back-up would be elegant, particularly since the quantitative aspect of the paper is one of its strongest assets. Low-hanging fruit for instance the correlation between DO and discharge from Fig. 1A, test of data in Fig. 2A (and add error bars reflect xxx in caption).

- We have added clarification in the caption for Fig. 2A to identify the error bars as standard deviation.
- As discussed in more detail above, we have added a multiple regression to the manuscript to highlight the factors that predict the magnitude in observed DO decreases.

- Suppl. Table 2: would be helpful to add coordinates of pools

- Coordinates for the pools are now provided in the Methods (Lines 408-412). We have also added them onto a new map (Supplementary Figure 1) and a new table (Supplementary Table 11).

- Suppl. Table 3: elegant to add K-value and df

- We have added the Chi-square statistic and degrees of freedom for the elements in Supplementary Table 3.

- Suppl. Table 6: give location

- We have modified the table heading to explicitly state that the parameters were measured during a flood pulse at New Mara Bridge. We have also modified the table headings for Supplementary Tables 7 and 8.

- Reference list: remove capital letters

- We have reformatted two citations that were in all capital letters, references 4 and 5.

- Reference 35 and 49 seem the same

- You are correct. We have removed reference 49 and replaced it with the original reference number, 35.

Reviewers' comments:

Reviewer #1 (Remarks to the Author):

Review of Dutton et al. Organic matter loading by hippopotami causes subsidy overload resulting in downstream hypoxia and fish kills

This paper has significant improvements compared to the first version. I also receive well the responses to the other reviewers, especially with respect to statistical analysis and more appropriate language/additional information in the abstract, introduction, and discussion. As a result, I am more convinced that the hippo pools can contribute to low DO water flushes and a bit more comfortable with the conclusions.

Despite the improvements, I still, however, find the discussion about reference states and fish biodiversity to be superficial. Relevant to that, I miss a good description of the local/regional context in the upper Mara River and in Kenya and the importance of this study to Kenyan situation and larger issues of conservation and savanna systems. The latter comment arises more clearly to me on this read due to the authors' observation that the river is so incised in the Masai Mara park.

I explain further with the following points:

1. The authors state that low DO events cause a reduction in fish diversity. This can be true in other systems (mostly temperate systems cited by the authors); however, at best there are weak or speculative data from the Mara river support this, and tropical systems can be different. In an evolutionary time-scale, it is likely that episodic events of low DO were somewhat common and occurring seasonally (i.e. during the start of wet seasons) and in many tropical floodplain rivers. This is likely stimulates speciation of fish adapted to these conditions, even if large fish kills occur during these events. This is a different kind of pressure and a different kind of effect than the anthropogenic impacts we observe recently, which have come on relatively rapidly and are (perhaps) less predictably episodic in nature. Hence, I'm still not fully comfortable with the authors' treatment of this concept. In addition to the unclear scientific explanation, I worry, like the reviewer 3's original comments, that the findings could be taken out of context and used in such a way that is bad for Kenyan conservation efforts. It could be useful for the authors to separate short-term dynamics from evolutionary time scales in the discussion, or, as there are not sufficient data for the Mara, leave it out or state more generally "affect fish community structure."

2. The authors state that the study area has deeply incised river banks within the Masai Mara Park, which reduce the floodplain-river connections and reduces the possibility that floodplain connections contribute low DO events. Why is the river incised, and how, long has this been occurring? It's likely on the scale of decades given the 3 – 6 m incision. Is it possible that the hippos have a stronger effect now on DO than they did historically? And if so, what does this mean for reference states? Upstream of the Park, the Mara basin has experienced deforestation, and increases in agriculture and urban land uses. Overall flow from the Mara River has been reduced along with the deforestation of the Mau forest (and perhaps climate change). Do these changes have anything to do with the lack of floodplain river connections and the relative contribution of the Hippos? I understand that these questions are at present beyond the scope of this study, but I again in general am finding it difficult to put this cool ecological phenomenon in a larger context. In short, I'm still searching for a well-developed answer to the "so what" question?

Some specific comments (the more important science/content comments are #6, #7, #13, #17, #19; others are mostly editorial):

1. Line 45, do not use quotes around the word natural as it makes it unclear what is really meant.

2. Line 64: add "to" before the word protect
3. Line 100: Comma before the word "and"
4. Line 111: The word "these" is confusing (what is it referring to?) In fact lines 111-115 are somewhat awkwardly written, and I had to read several times.
5. Line 191: Why three-fold?
6. Lines 193, and lines 199 – 203: If it takes only 11% of HPW to be entrained to cause hypoxia, and there is 65% of HPW at the NMB site, wouldn't hypoxic events occur a lot more often they do? I wonder if some model assumptions need to be adjusted, or this statement explained better.
7. Line 241 (and the Title): the word "overload" seems too much like a value-loaded word to me and is not defined (what is an overload?) This is also inconsistent if the argument is that hypoxic occur events more often in a reference state when hippos would have been more common. Would prefer a different word.
8. Line 246: Insert the word "by" before "upstream".
9. Line 250 – 251: not sure what "this" refers to.
10. Line 298: Delete the phrase "And thus are often considered desirable" (too value-laded, and no references are given to support this)
11. Line 299: Delete the word "degrade" – also too value-laden
12. Line 313: Would it help clarify further after "productivity" to add the phrase: "in the whole river system" ?
13. Lines 320-225 contain some of the discussion that am reacting to in my comments above. Consider qualifying the discussion in terms of short-term effects vs. longer term or evolutionary effects. Also, might want to consider adding to the caveat about human effects that they may not be so predictable or episodic in nature.
14. Line 385: re-word: "...beginning DO data, which were....." (Not DO was)
15. Line 388: I was confused by the phrase "that resulted in a decrease in DO" Maybe just delete.
16. Line 392 – 393: Incorrect name. It should be the Government Chemist of Kenya, not the Kenya Government Chemist.
17. Line 398: In several places in the document the authors state that increases in methane, hydrogen sulphide, and ammonium increase downstream during flushing events, however no data are presented for the gases and no references given. Later in the paper, it seems that NH₄ concentration wasn't always fully in sync with low DO.
18. Line 533: Phrase "peak drop" is a bit confusing at first; consider a different phrase. Related to that, Figure 2 would be more intuitive if the axis were arranged on a negative scale, so the lines go down, not up, to describe a "drop."
19. I think I understand from the event descriptions that NH₄ and other water quality constituents didn't always follow the same relationship with DO in each event. This is pretty interesting (but contradicts a bit the authors' earlier statement, see comment 17). It would be helpful and interesting to include graphs of the NH₄ (and maybe others) along with the storm event graphs in the supplementary information (Supplementary Figures 4 – 6).
20. Figure 2: Would be good to adjust the Y axis scales on panels b and c so that they have the same increment.

One final note: The author-list is 100% American scientists and no Kenyans. I understand that funding constraints may prevent as much collaboration with foreign scientists as one might like, but it is a pity that Kenyan scientists are not included in this study, as there are many there, and we in should try to include more under-resourced regions of the world in environmental science. There are many Kenyan scientists, and the country, the parks, their conservation efforts, would benefit greatly from being included in a study like this.

Reviewer #2 (Remarks to the Author):

I was very interested to review this revised version of a manuscript describing hippo impacts on a river ecosystem in Africa. I appreciate the authors' extensive attention to previous reviewer input and suggestions. Addition of new analyses and their interpretation is valuable, and strengthen the manuscript, which uses an impressive array of measurements and provides novel information that will be of broad interest. I believe that this has resulted in improvements that move the paper substantially closer to a level that would warrant published in this journal. However, some work remains, in particular to provide more clarity in the presentation of the manuscript, and to better support its main conclusions, as described below in more detail.

I see three major areas to address. First, the manuscript is not quite as succinct as necessary for Nature Communications; this is made challenging by the diverse results presented in support of the central ideas of the manuscript. The text is difficult to follow in places, but some minor editing and streamlining can address this.

Second, while the analyses are now stronger and more convincing, they as yet don't provide as clear and complete picture of the system dynamics as is needed. Estimates that relate the volume of hypoxic pool water in the channel before storms to oxygen decreases observed during floods represent important and valuable information (lines 193-202). These estimates were made for an "average flushing flow" condition. Such conditions actually represent very small increases in flow (2x baseflow) for a river system, where flows rapidly increase 10-100x and greater during storms. After accounting for other variables (e.g. reaeration, etc.) I would be interested to see how much of the observed oxygen depletion (the integrated total oxygen deficit observed for entire storms) may be explained by the contribution of Hippo influenced pool water, across the entire range of floods (i.e. those presented in Figure 1b). Using the information gathered, the authors can relate climate variability (i.e. storm size and frequency) to oxygen sags to gain the information needed to put these results into context of hydrologic variability in the river (also see next comment, regarding human impacts on river flow). This seems to me to be a more synthetic, robust and straightforward way to present these key data, relative to Fig. 1b, which is difficult to interpret. A related issue here is that while the newly added analyses of flood O₂ data (presented starting on line 111) are helpful, they are not clearly presented, and some important details of the statistical model selection are missing, preventing full evaluation. For example, because the volume of hypoxic pool water is a fixed amount, one might expect that the influence of hippo pools on total river oxygen deficits during storms would decline with total flood volume (storm size), yet this parameter is not analysed for, at least as far as I can tell.

Third, given the role of hydrologic variables (i.e. peak flow, antecedent flow, runoff volume) in observed river hypoxia that begin to emerge from the multivariate analyses (starting line 111), some additional information about river hydrology is needed. The Mara River is only briefly described as a "relatively well-protected river". This requires more explanation, especially in the context of one of the major conclusions of the study (lines 33-32, 345- 348) regarding the prevalence of hypoxia in tropical rivers. There appear to be substantial human influence on land cover in the headwaters of the river - a quick (and by now mean exhaustive) scan of recent work show a number of studies that address land use impacts on hydrology and river flow change in the catchment (e.g. Mango et al. 2011, Mati 2008, Mwangi et al. 2016). Integration of this information to help determine how much (if any) of the observed hypoxia may have been influenced by interactions with human changes to river flow regime seems absolutely essential to better defend the conclusions regarding the role of hypoxia as a natural feature of tropical river ecosystems.

Two minor comments:

The abstract states that 49 hypoxic events were observed, but subsequent text (line 102) indicates that thirteen of the flushing flows resulted in hypoxia. Please clarify.

Hippo pool water appears to be enriched in SRP/PO₄ (table s9), as expected. However the SRP levels during floods are quite low (tables s6-8). A minor point but one that perhaps requires some explanation in the supplement.

Reviewer #3 (Remarks to the Author):

I have carefully read the revised manuscript and the author's replies to the reviewer comments. I think that the revised manuscript has improved in clarity and the findings of the study have been interpreted in a broader context, by referring to what aquatic systems may have looked like in the present of now-extinct, or seriously reduced in abundance, megafauna. With this context the paper is an important contribution to our conceptual thinking about what pristine aquatic ecosystems would have looked like. This relates both to past ecosystems from a paleo-ecological perspective, as well as present ecosystem subject to defaunation and future ecosystems subject to rewilding.

Responses to reviewers are bulleted inline below. Relevant changes in the manuscript are highlighted.

Reviewers' comments:

Reviewer #1 (Remarks to the Author):

Review of Dutton et al. Organic matter loading by hippopotami causes subsidy overload resulting in downstream hypoxia and fish kills

This paper has significant improvements compared to the first version. I also receive well the responses to the other reviewers, especially with respect to statistical analysis and more appropriate language/additional information in the abstract, introduction, and discussion. As a result, I am more convinced that the hippo pools can contribute to low DO water flushes and a bit more comfortable with the conclusions.

Despite the improvements, I still, however, find the discussion about reference states and fish biodiversity to be superficial. Relevant to that, I miss a good description of the local/regional context in the upper Mara River and in Kenya and the importance of this study to Kenyan situation and larger issues of conservation and savanna systems. The latter comment arises more clearly to me on this read due to the authors' observation that the river is so incised in the Masai Mara park.

- Thank you for taking the time to review our manuscript. We appreciate the feedback you provided. We have addressed your comments below.

I explain further with the following points:

1. The authors state that low DO events cause a reduction in fish diversity. This can be true in other systems (mostly temperate systems cited by the authors); however, at best there are weak or speculative data from the Mara river support this, and tropical systems can be different. In an evolutionary time-scale, it is likely that episodic events of low DO were somewhat common and occurring seasonally (i.e. during the start of wet seasons) and in many tropical floodplain rivers. This is likely stimulates speciation of fish adapted to these conditions, even if large fish kills occur during these events. This is a different kind of pressure and a different kind of effect than the anthropogenic impacts we observe recently, which have come on relatively rapidly and are (perhaps) less predictably episodic in nature. Hence, I'm still not fully comfortable with the authors' treatment of this concept. In addition to the unclear scientific explanation, I worry, like the reviewer 3's original comments, that the findings could be taken out of context and used in such a way that is bad for Kenyan conservation efforts. It could be useful for the authors to separate short-term dynamics from evolutionary time scales in the discussion, or, as there are not sufficient data for the Mara, leave it out or state more generally "affect fish community structure."

- We acknowledge your point about low DO events possibly having positive effects on aquatic diversity, particularly in systems where species have evolved in response to these events. Where

we referred to high levels of loading decreasing diversity (Line 42), we were referring to the majority of systems where this has been studied, as you noted given our citations. However, we have now revised this to read “potential loss of diversity” and added an additional citation (Isbell et al., 2013), which talks specifically about the loss of diversity due to high loading of nutrients (Lines 41-44).

- We also appreciate your point about evolutionary responses to frequent hypoxic events leading to speciation and potential increase in diversity. As suggested, we have included a brief discussion of possible diversity responses to these events both in the short-term and at evolutionary time scales (Lines 314-325)
- In regards to taking these findings out of context, we included text in our last revision addressing this issue, however, we have now expanded upon it in the current manuscript. It now reads, “However, great caution should be used in applying this understanding to current incidences of hypoxia due to anthropogenic loading, which may differ in frequency and periodicity from natural events, occur in ecosystems where organisms have not evolved to withstand this stressor, and where pollution sources may carry additional contaminants with other negative consequences^{51,52}.” (Lines 348–353)

2. The authors state that the study area has deeply incised river banks within the Masai Mara Park, which reduce the floodplain-river connections and reduces the possibility that floodplain connections contribute low DO events. Why is the river incised, and how, long has this been occurring? It’s likely on the scale of decades given the 3 – 6 m incision. Is it possible that the hippos have a stronger effect now on DO than they did historically? And if so, what does this mean for reference states? Upstream of the Park, the Mara basin has experienced deforestation, and increases in agriculture and urban land uses. Overall flow from the Mara River has been reduced along with the deforestation of the Mau forest (and perhaps climate change). Do these changes have anything to do with the lack of flood-plain river connections and the relative contribution of the Hippos? I understand that these questions are at present beyond the scope of this study, but I again in general am finding it difficult to put this cool ecological phenomenon in a larger context. In short, I’m still searching for a well-developed answer to the “so what” question?

- A recent study has documented the incision of the Mara River in the vicinity of the study site and the consequent isolation of the river from its floodplain (McClain et al. 2014), and others in the region have shown that this incision is not clearly related to catchment land use (Miller and Doyle 2014). While the Mara River channel incision could be a relatively recent development, this kind of channel geomorphology is not unique. Sequences of deeper pools along dryland river channels are known as “waterholes” in Australia and southern Africa, where geomorphological studies have been conducted by Gerald Nanson and others (e.g., Nanson et al. 2005). Channel incision is not necessary for waterholes to exist; they are believed to reflect geomorphological control points such as bedrock or hardpan outcrops that impound water and force scouring by focused flow during high discharge events. While all of this is too much extraneous detail to put in our manuscript, we believe that our observations are not unique to the Mara River system and likely apply to many other river systems throughout the wet-dry tropics of Africa and elsewhere. We have now included references to these papers and noted that this geomorphology is fairly typical of rivers in this region in Lines 81-84.

- We have also added an additional paragraph in the discussion about changes in the upper catchment of the basin and the changes in the hippopotamus populations. We also provide an explicit caveat that, “Our research supports the hypothesis that flushing of hippo pools is sufficient to cause hypoxic events in the river, but it does not preclude the additional influence of other anthropogenic factors. Future research in the Mara should continue to investigate the relative contributions of anthropogenic and natural drivers in ecosystem dynamics of this river.” (Lines 355-368)
- Our manuscript is novel in that we show the mechanisms behind how a large mammal can cause repeated hypoxic events in a large, free-flowing river. This has significant implications for our understanding about reference states in aquatic systems prior to the extirpation of megafauna (Lines 337-353, 370-381).

Some specific comments (the more important science/content comments are #6, #7, #13, #17, #19; others are mostly editorial):

1. Line 45, do not use quotes around the word natural as it makes it unclear what is really meant.

- Removed quotes.

2. Line 64: add “to” before the word protect

- Done.

3. Line 100: Comma before the word “and”

- Done.

4. Line 111: The word “these” is confusing (what is it referring to?) In fact lines 111-115 are somewhat awkwardly written, and I had to read several times.

- “These” was referring to the in-situ data mentioned in the prior paragraph. We have removed the term “these” to clear up any confusion. The sentence now reads, “We used in situ data to investigate whether the degree of hypoxia resulting from a flushing flow was affected by how much time the pools had to become anoxic since the last flushing (using time since previous flushing and initial DO), how fast the pools were flushed (time to peak discharge), and the influence of discharge on entrainment and dilution of the anoxic hippo pool water (initial and peak discharge).” (Lines 116-120)

5. Line 191: Why three-fold?

- We have added additional text to emphasize that the three-fold increase in DO drop is due to the increased depth in the river. The text now reads, “Using a modeling approach to extrapolate the results from experimental streams with 15-cm depth to a river channel with 100-cm depth (similar to the mainstem Mara River), and assuming the same reaeration rates (see Methods), we estimate that the maximum DO drop in the river would be three-fold greater than in the experimental channels because of the increased depth in the river channel.” (Lines 192-196)

6. Lines 193, and lines 199 – 203: If it takes only 11% of HPW to be entrained to cause hypoxia, and there is 65% of HPW at the NMB site, wouldn't hypoxic events occur a lot more often they do? I wonder if some model assumptions need to be adjusted, or this statement explained better.

- We have modified that statement to make it more clear and added text to explain why there is more variation than expected given the modeling results.
 - “Assuming all hippo pools were flushed and had identical BOD as the HPW used in the experimental stream addition (see Supplementary Table 3), and integrating under an average flushing flow hydrograph, we estimate that water from hippo pools could contribute up to 65% of the total volume of water moving past the NMB site during the average flushing flow, suggesting there is more than enough HPW to drive the Mara River to hypoxia.” (Lines 204-209)
 - “However, it is unlikely that all hippo pools would be flushed during the same flushing flow and that their HPW would all have the same BOD as the HPW used in the model and experimental stream addition (see Supplementary Table 3). BOD will likely be higher in hippo pools with greater loading and less flushing, and in hippo pools that have not experienced a recent flushing flow. Differences in BOD and its constituents among hippo pools and variation in flushing flows over space and time among the 171 hippo pools in the Mara River and its tributaries likely explain some of the unexplained variability in the response of DO at the NMB during different flushing flow events.” (Lines 287-295)

7. Line 241 (and the Title): the word “overload” seems too much like a value-loaded word to me and is not defined (what is an overload?) This is also inconsistent if the argument is that hypoxic occur events more often in a reference state when hippos would have been more common. Would prefer a different word.

- We believe “overload” is the best term to describe the situation. A critical load represents the amount of organic matter or nutrients that a system can safely absorb before there is a change in ecosystem state (Groffman et al, 2006). An overload is loading in excess of that critical load. To be more clear, we have now defined it in the text.
 - “Higher levels of loading can lead to eutrophication, hypoxia, potential loss of diversity, and altered ecosystem functioning⁵⁻⁹. Loading of organic matter and nutrients above a critical threshold results in an overload that switches the system from an aerobic to an anaerobic state¹⁰. (Lines 41-44).
- We don't believe calling these events a subsidy overload is inconsistent with the argument that hypoxic events potentially occurred more often in reference states, as the definition of critical load (and thus our use of the term overload) refers to a biogeochemical threshold that could be reached in natural or impacted systems.

8. Line 246: Insert the word “by” before “upstream”.

- Inserted.

9. Line 250 – 251: not sure what “this” refers to.

- Changed from “this” to “the”. (Lines 253-256)

- “During flushing flows, downstream reaches experience an immediate and rapid decrease in DO attributable to both mixing and oxygen consumption processes, although the effect is eventually diminished as the water moves downstream and DO returns to normal through reaeration. “

10. Line 298: Delete the phrase “And thus are often considered desirable” (too value-laded, and no references are given to support this)

- We have deleted this phrase.

11. Line 299: Delete the word “degrade” – also too value-laden

- Changed to “alter”.

12. Line 313: Would it help clarify further after “productivity” to add the phrase: “in the whole river system” ?

- Changed.

13. Lines 320-225 contain some of the discussion that am reacting to in my comments above. Consider qualifying the discussion in terms of short-term effects vs. longer term or evolutionary effects. Also, might want to consider adding to the caveat about human effects that they may not be so predictable or episodic in nature.

- As noted above, we have included a brief discussion of possible diversity responses to these events both in the short-term and at evolutionary time scales (Lines 314-325). We have now added an additional caveat about how human effects may differ from natural effects. The phrase now states:
 - “However, great caution should be used in applying this understanding to current incidences of hypoxia due to anthropogenic loading, which may differ in frequency and periodicity from natural events, occur in ecosystems where organisms have not evolved to withstand this stressor, and where pollution sources may carry additional contaminants with other negative consequences^{51,52}.” (Lines 348-353)

14. Line 385: re-word: “...beginning DO data, which were.....” (Not DO was)

- Changed.

15. Line 388: I was confused by the phrase “that resulted in a decrease in DO” Maybe just delete.

- Deleted.

16. Line 392 – 393: Incorrect name. It should be the Government Chemist of Kenya, not the Kenya Government Chemist.

- Changed.

17. Line 398: In several places in the document the authors state that increases in methane, hydrogen sulphide, and ammonium increase downstream during flushing events, however no data are presented for the gases and no references given. Later in the paper, it seems that NH₄ concentration wasn't always fully in sync with low DO.

- We have clarified in the text that our knowledge about the high concentrations of hydrogen sulfide comes from the samples taken from hippopotamus pools (Supplementary Table 3). (Lines 427-429).
- We no longer refer to possible toxicity of ammonia (which would be pH dependent and is thus difficult to evaluate), or methane.
- We would not necessarily expect NH_4^+ to vary inversely with DO in this system, particularly because nitrification would quickly consume excess NH_4^+ with even modest oxygenation. We have added a discussion about this in the supplementary file (Supplementary Note 1).

18. Line 533: Phrase “peak drop” is a bit confusing at first; consider a different phrase. Related to that, Figure 2 would be more intuitive if the axis were arranged on a negative scale, so the lines go down, not up, to describe a “drop.”

- As suggested, the axis on Figure 2 has been rearranged. It is now arranged on a negative scale.

19. I think I understand from the event descriptions that NH_4 and other water quality constituents didn't always follow the same relationship with DO in each event. This is pretty interesting (but contradicts a bit the authors' earlier statement, see comment 17). It would be helpful and interesting to include graphs of the NH_4 (and maybe others) along with the storm event graphs in the supplementary information (Supplementary Figures 4 – 6).

- That is correct, the water quality constituents that we measured during the three flushing events did not always increase together (see above comment in reference to NH_4^+). We do know that hippo pools can have high levels of ammonium, methane and hydrogen sulfide (Supplementary Table 3). We also know that these constituents get flushed out of the hippo pools during flushing flows. The degree that they do get flushed and their impact on dissolved oxygen downstream are difficult to discern. We have addressed this now in the text in Lines 288-295.
- We were not able to measure methane and hydrogen sulfide during these three flushing events because they were captured with automatic samplers that were later retrieved. We would not have been able to preserve the samples for the analysis of methane and hydrogen sulfide, which are subject to degassing and oxidation.
- We have added graphs of NH_4^+ and specific conductivity to Supplementary Figures 4-6.

20. Figure 2: Would be good to adjust the Y axis scales on panels b and c so that they have the same increment.

- As suggested, we have modified the figures so that the Y axis scale is the same in both Figures 2b and 2c.

One final note: The author-list is 100% American scientists and no Kenyans. I understand that funding constraints may prevent as much collaboration with foreign scientists as one might like, but it is a pity that Kenyan scientists are not included in this study, as there are many there, and we should try to include more under-resourced regions of the world in environmental science. There are many Kenyan scientists, and the country, the parks, their conservation efforts, would benefit greatly from being included in a study like this.

- Thank you for this very important point. We do have many wonderful colleagues throughout Kenya, Tanzania and Rwanda with whom we are collaborating on studies and scientific papers. One manuscript is just being published and several others are in the preparation stage. This paper resulted from work conducted by a PhD student (Christopher Dutton) for his dissertation at Yale University. His initial research in Kenya was funded by a US National Science Foundation-funded grant on the role of subsidies in the Mara River (DEB 1354053 and 1354062), and this work was conducted on the side of that larger work. The two Kenyan colleagues with whom we could have worked with on the development of this manuscript passed away several years ago during the very early stages of this research. Our current close collaborators in Kenya are involved in researching food webs and play an important role in the larger NSF grant and other subsequent work. However, they did not contribute to the work conducted as part of this manuscript. All authors on this manuscript contributed equally and were absolutely essential to the collection, interpretation and disentanglement of this fascinating phenomenon.
- We continue to seek opportunities to involve more local scientists in our work. As part of that, we have hosted courses in Kenya over the last two years and brought local scientists out to these study sites to form future collaborations.
 - <https://mara.yale.edu/news/food-web-short-course>
 - <https://mara.yale.edu/food-web-short-course>
 - <https://mara.yale.edu/news/nutrient-uptake-and-metabolism-short-course>
- We have also recently taught courses in Kenya and Tanzania to local scientists on how to develop and build their own low-cost sensors for environment monitoring similar to the ones we use in our study.
 - <https://mara.yale.edu/low-cost-sensor-course-tanzania>
- Additionally, we have taught multiple courses in Tanzania and Rwanda on suspended sediment transport and continue to work with scientists from there on manuscripts from that work.
- Therefore, we feel that we have played a significant role in building local scientific capacity, even though regrettably we do not have Kenyan co-authors on this paper.

Reviewer #2 (Remarks to the Author):

I was very interested to review this revised version of a manuscript describing hippo impacts on a river ecosystem in Africa. I appreciate the authors' extensive attention to previous reviewer input and suggestions. Addition of new analyses and their interpretation is valuable, and strengthen the manuscript, which uses an impressive array of measurements and provides novel information that will be of broad interest. I believe that this has resulted in improvements that move the paper substantially closer to a level that would warrant published in this journal. However, some work remains, in particular to provide more clarity in the presentation of the manuscript, and to better support its main conclusions, as described below in more detail.

- Thank you for taking the time to review our manuscript. We appreciate the feedback you provided.

I see three major areas to address. First, the manuscript is not quite as succinct as necessary for Nature Communications; this is made challenging by the diverse results presented in support of the central

ideas of the manuscript. The text is difficult to follow in places, but some minor editing and streamlining can address this.

- We have made many modifications throughout the text to streamline it and make it more clear for the reader. All of these modifications have been made in direct response to the three reviewer comments, detailed below.

Second, while the analyses are now stronger and more convincing, they as yet don't provide as clear and complete picture of the system dynamics as is needed. Estimates that relate the volume of hypoxic pool water in the channel before storms to oxygen decreases observed during floods represent important and valuable information (lines 193-202). These estimates were made for an "average flushing flow" condition. Such conditions actually represent very small increases in flow (2x baseflow) for a river system, where flows rapidly increase 10-100x and greater during storms. After accounting for other variables (e.g. reaeration, etc.) I would be interested to see how much of the observed oxygen depletion (the integrated total oxygen deficit observed for entire storms) may be explained by the contribution of Hippo influenced pool water, across the entire range of floods (i.e. those presented in Figure 1b). Using the information gathered, the authors can relate climate variability (i.e. storm size and frequency) to oxygen sags to gain the information needed to put these results into context of hydrologic variability in the river (also see next comment, regarding human impacts on river flow). This seems to me to be a more synthetic, robust and straightforward way to present these key data, relative to Fig. 1b, which is difficult to interpret.

- We agree that this would be interesting and we considered attempting it, but concluded that we do not possess the data that would be required for such an analysis. This should be based on a full hydrodynamic model of the river hydrology incorporating the spatial provenance of flushing flows, pool-specific BOD and volume estimates, transit times for flow through the network, changes in reaeration rates along the channel network and as a function of flow and water level, and consideration of the duration and shape of flood pulses as well as their maximum discharge. We provide a discussion on this point and identify it as a topic for future research (Lines 287-307).
- We also provide additional discussion on why it is difficult to generalize about the response of DO to the flushing of hippo pools using the long-term data from the dissolved oxygen logger at the NMB site. This now reads,
 - "However, it is unlikely that all hippo pools would be flushed during the same flushing flow and that their HPW would all have the same BOD as the HPW used in the model and experimental stream addition (see Supplementary Table 9). BOD will likely be higher in hippo pools with greater loading and less discharge, and in hippo pools that have not experienced a recent flushing flow. Differences in BOD and its constituents among hippo pools and variation in flushing flows over space and time among the 171 hippo pools in the Mara River and its tributaries likely explain some of the unexplained variability in the response of DO at the NMB during different flushing flow events (Fig. 1b)." (Lines 287-295)
- The "average flushing flow" we used for this analysis is a much more frequent occurrence in this system than very large floods. We have added additional text in the methods to clarify this point.

- “Peak discharge ranged from 6 to 197 m³ s⁻¹, and 43 out of 49 of the flushing flows had a peak discharge less than 65 m³ s⁻¹. The average flushing flow increased three-fold over the calculated baseflow.” (Lines 408-410)
- Figure 1b shows the magnitude of dissolved oxygen drop in relation to the flushing flow peak for all 49 of the flushing flows that resulted in a decrease in dissolved oxygen. This information is very important for putting the magnitude of the drop in dissolved oxygen in context to how quickly it occurred during the flushing flows.

A related issue here is that while the newly added analyses of flood O₂ data (presented starting on line 111) are helpful, they are not clearly presented, and some important details of the statistical model selection are missing, preventing full evaluation. For example, because the volume of hypoxic pool water is a fixed amount, one might expect that the influence of hippo pools on total river oxygen deficits during storms would decline with total flood volume (storm size), yet this parameter is not analysed for, at least as far as I can tell.

- We did not include total storm size in our model. However, we did include peak discharge in our model, which is correlated to total storm size. We believe that peak discharge is a more appropriate term to utilize in our model since we also include the time to peak discharge and beginning discharge as additional variables. All three of our terms are related to the total size of the storm but provide additional information about the dynamics of the event and are independent of one another, which is an important assumption of the analysis. We now explicitly state this in the text. (Lines 418-420)
 - “We did not include the total storm size in the model because the terms peak discharge, initial discharge and time to peak discharge all account for the total storm size”
- Yes, we agree that the largest floods would make any effects of hippo pool water inconsequential because of dilution of the fixed volume of hippo pool water. However, we think the more frequent, smaller flushing flows are important over the course of the year, and it is those floods that we analyze in this study. We have added additional information in the text to highlight that out of all 49 flushing flows that resulted in a decrease in DO, the majority of them (43) had a peak discharge less than 65 m³ s⁻¹ (less than an approximate 4-fold increase over average baseflow). (Lines 408-410)
- We have edited the text about the model selection to make the explanation clearer (Lines 116-120). In the last revision, we also included additional information in the methods (Lines 412-423).

Third, given the role of hydrologic variables (i.e. peak flow, antecedent flow, runoff volume) in observed river hypoxia that begin to emerge from the multivariate analyses (starting line 111), some additional information about river hydrology is needed. The Mara River is only briefly described as a “relatively well-protected river”. This requires more explanation, especially in the context of one of the major conclusions of the study (lines 33-32, 345- 348) regarding the prevalence of hypoxia in tropical rivers. There appear to be substantial human influence on land cover in the headwaters of the river- a quick (and by now mean exhaustive) scan of recent work show a number of studies that address land use impacts on hydrology and river flow change in the catchment (e.g. Mango et al. 2011, Mati 2008, Mwangi et al. 2016). Integration of this information to help determine how much (if any) of the observed hypoxia may have been influenced by interactions with human changes to

river flow regime seems absolutely essential to better defend the conclusions regarding the role of hypoxia as a natural feature of tropical river ecosystems.

- We have added additional discussion concerning this in Lines 355-368.
 - “Although the middle reaches of the Mara River are relatively well-protected, current patterns of land use change and development in the upper Mara River Basin have been cited as influencing hydrology in the Mara River^{42,53-56}. Additionally, the hippopotamus population in the middle reaches of the Mara River has recently stabilized after a 1500% increase since the first surveys conducted in the 1950’s^{19,40,57}. The size of pre-colonial populations of hippopotami in the Mara is not known, although globally their current range is a fraction of their historical range due to habitat loss and extirpation of this species by humans^{30,35}. There remain open questions about the degree to which changes in the upper basin interact with changing nutrient and organic loading from hippopotami to influence long-term river ecosystem dynamics. Our research supports the hypothesis that flushing of hippo pools is sufficient to cause hypoxic events in the river, but it does not preclude the additional influence of other anthropogenic factors. Future research in the Mara should continue to investigate the relative contributions of anthropogenic and natural drivers in ecosystem dynamics of this river.”

Two minor comments:

The abstract states that 49 hypoxic events were observed, but subsequent text (line 102) indicates that thirteen of the flushing flows resulted in hypoxia. Please clarify.

- We have changed the text in the abstract to read, “We documented 49 high flow events over three years that caused dissolved oxygen decreases, including 13 events resulting in hypoxia, and 9 fish kills over five years.

Hippo pool water appears to be enriched in SRP/PO₄ (table s9), as expected. However the SRP levels during floods are quite low (tables s6-8). A minor point but one that perhaps requires some explanation in the supplement.

- We agree that there seems to be an unexplained SRP sink, but we lack the information to offer an explanation. One possibility is that iron precipitation is removing the SRP by sorption to Fe oxyhydroxides, but we have no measurements to assess that. We have added a Supplementary Note 1 in the supplementary information file to address this possibility, which may be of interest to a subset of readers.

Reviewer #3 (Remarks to the Author):

I have carefully read the revised manuscript and the author's replies to the reviewer comments. I think that the revised manuscript has improved in clarity and the findings of the study have been interpreted in a broader context, by referring to what aquatic systems may have looked like in the present of now-extinct, or seriously reduced in abundance, megafauna. With this context the paper is an important contribution to our conceptual thinking about what pristine aquatic ecosystems would have looked like.

This relates both to past ecosystems from a paleo-ecological perspective, as well as present ecosystem subject to defaunation and future ecosystems subject to rewilding.

- Thank you for taking the time to review our manuscript. We appreciate the feedback you provided.

REVIEWERS' COMMENTS:

Reviewer #2 (Remarks to the Author):

This paper should ultimately be published in Nature Communications, but there remain several areas to address to bring it to the standard of the journal, and to ensure its accessibility to a wide audience. Overall, the responses to the previous reviewer comments are reasonable and have improved many areas of the manuscript. While I am not completely satisfied with all of them, I am willing to accept most at this point, but do see three points to address:

First, while the paper brings together an impressive weight of evidence, it presents a problem in that its presentation is not succinct. In particular, while the new text is helpful, in places it is not completely integrated with previous content. Some streamlining is necessary, and if possible, reductions in the overall length of the main body would be beneficial. For example, the paragraph starting on 359 wanders, starting with a discussion of hydrology but also including sentences about hippo population change that would fit better elsewhere (e.g. the previous paragraph). I suggest a revision of this paragraph to focus more directly and specifically on the hydrologic changes in the watershed. Generic human influences on hydrology are noted (line 361) but no indication is given about what these changes are, or how they might have interacted with hippo pool flushing.

I remain troubled by the overstatement in the last part of the final sentence of the abstract. Earlier reviewer comments still seem relevant. Even though it is qualified later on, this statement (line 34-35) is simply too strong and general, especially given the frequency of hypoxia observed in other natural stream systems.

Third, a major premise of this paper is that hypoxia is rare in natural settings. Being unfamiliar with the literature, I made a quick scan. Following this, I now wonder if this point is slightly overstated in the paper. Hypoxia under low flow is well known in headwater streams (Lake 2000) and intermittent rivers (Daltry et al. 2014 Biosci.) of arid areas. The authors have cited some Austral and Amazonian examples (citations 11-14) of river hypoxia but there are certainly others that may be useful to consider (e.g. Sargent et al. 2017 Ecosphere, Small 2014 Plos, Cech 1990, Hladyz 2011 J. Hydrology) including some African ones (see below). While somewhat rare, depending on climate and watershed cover (wetlands), hypoxic events are perhaps more common than indicated in the ms; some minor revisions may be necessary. In any case, the work of Lauren Chapman should be cited in here. A good place to start would be a chapter on low oxygen fishes in Riesch et al's 2015 book, which includes some African data. It seems important to cite some of this work around line 325 as well.

Minor points

In the paragraph starting on line 116, I wonder if a table summarizing these statistics would be easier to digest. This is a little hard to follow.

Line 424 I am not yet convinced this is true. It would be nice to back this up statistically.

Responses to reviewers are bulleted inline below. Relevant changes in the manuscript are in track changes.

REVIEWERS' COMMENTS:

Reviewer #2 (Remarks to the Author):

This paper should ultimately be published in Nature Communications, but there remain several areas to address to bring it to the standard of the journal, and to ensure its accessibility to a wide audience. Overall, the responses to the previous reviewer comments are reasonable and have improved many areas of the manuscript. While I am not completely satisfied with all of them, I am willing to accept most at this point, but do see three points to address:

- Thank you very much for taking the time to provide such helpful comments throughout this process.

First, while the paper brings together an impressive weight of evidence, it presents a problem in that its presentation is not succinct. In particular, while the new text is helpful, in places it is not completely integrated with previous content. Some streamlining is necessary, and if possible, reductions in the overall length of the main body would be beneficial. For example, the paragraph starting on 359 wanders, starting with a discussion of hydrology but also including sentences about hippo population change that would fit better elsewhere (e.g. the previous paragraph). I suggest a revision of this paragraph to focus more directly and specifically on the hydrologic changes in the watershed. Generic human influences on hydrology are noted (line 361) but no indication is given about what these changes are, or how they might have interacted with hippo pool flushing.

- We are hesitant to reduce much of the text at this point, as much of it was suggested during the review process, and our manuscript is still within the length requirements of the journal. However, we have streamlined sections where possible. Additionally, we have moved many of the details from the methods sections on reaeration modelling, sediment fingerprinting and HPW/feces collection into the supplementary information file (Supplementary Notes 2-4).
- Regarding the paragraph at line 358 (previously line 359) that the reviewer mentioned, our aim in this paragraph was to discuss additional factors potentially influencing hypoxic events in the Mara River, but about which we need more information. We have restructured this paragraph to clarify that open questions remain about the degree to which changes in the upper catchment and changes in the hippo population may contribute to these hypoxic flushing flows. We have also clarified what the possible changes in hydrology are (more extreme high and low flows), although we don't expound upon the implications of these changes in detail as there is currently no conclusive evidence they are occurring.

I remain troubled by the overstatement in the last part of the final sentence of the abstract. Earlier reviewer comments still seem relevant. Even though it is qualified later on, this statement (line 34-35) is simply too strong and general, especially given the frequency of hypoxia observed in other natural stream systems.

- We have toned down the last sentence in the abstract. It now says, “Frequent hypoxia may be a natural part of tropical river ecosystem function, particularly in rivers impacted by large wildlife.”

Third, a major premise of this paper is that hypoxia is rare in natural settings. Being unfamiliar with the literature, I made a quick scan. Following this, I now wonder if this point is slightly overstated in the paper. Hypoxia under low flow is well known in headwater streams (Lake 2000) and intermittent rivers (Daltry et al. 2014 Biosci.) of arid areas. The authors have cited some Austral and Amazonian examples (citations 11-14) of river hypoxia but there are certainly others that may be useful to consider (e.g. Sergent et al. 2017 Ecosphere, Small 2014 Plos, Cech 1990, Hladyz 2011 J. Hydrology) including some African ones (see below). While somewhat rare, depending on climate and watershed cover (wetlands), hypoxic events are perhaps more common than indicated in the ms; some minor revisions may be necessary. In any case, the work of Lauren Chapman should be cited in here. A good place to start would be a chapter on low oxygen fishes in Riesch et al’s 2015 book, which includes some African data. It seems important to cite some of this work around line 325 as well.

- There is no question that tropical waters are frequently oxygen depleted and that many species of tropical fishes display a number of adaptations to life in low-oxygen waters (as we note in Lines 48-49), but chronic oxygen depletion occurs mostly in more stagnant, often vegetated waters of floodplains and wetlands, rather than in open river channels like the Mara River. While chronic oxygen depletion selects for tolerant fish species, episodic and rapid oxygen depletion strongly impacts fishes and other aquatic biota that normally live with better oxygen availability, and thus represents a stress on the ecosystem. Unfortunately, comprehensive surveys of the occurrence of river hypoxia in tropical rivers are not available in the literature; the studies we have already cited are the best examples we know of that pertain to rivers of the scale of the Mara River.
- We have modified several sections in the text to more accurately reflect these ideas and incorporated several of your suggested citations as follows:
 - We have modified the line in the abstract that stated hypoxia was rare in unpolluted systems to emphasize that we are explicitly talking about rivers. The line now states, “Hypoxia is often attributed to anthropogenic pollution and is not common in unpolluted rivers.” (Lines 26-27).
 - We have also modified a line in the discussion and added an additional reference suggested by the reviewer stating that organic matter loading can also lead to a decrease in secondary production in certain instances. That line now reads, “In most other studies of terrestrial subsidies of organic matter and nutrients to aquatic ecosystems, the subsidies have been viewed as enhancing secondary production, including fisheries, although there are other instances

of high organic matter loading in which hypoxia and decreased secondary production can occur.” (Lines 328-331).

- We have added a citation to Chapman’s work in Lines 339-341. That line reads, “On evolutionary time scales, many species of fishes in tropical floodplain rivers may have evolved in response to frequent hypoxia and thus be adapted to withstand these events.”

Minor points

In the paragraph starting on line 116, I wonder if a table summarizing these statistics would be easier to digest. This is a little hard to follow.

- We have modified the text slightly to indicate that these statistics were from the use of a multiple linear regression. We have chosen to keep the statistics within the paragraph for the readers that prefer them in line.

Line 424 I am not yet convinced this is true. It would be nice to back this up statistically.

- We have rewritten this sentence with the rationale for not including the total storm size in our model. The sentence now states, “We did not include total storm size in the model; rather, we included initial discharge, peak discharge and time to peak discharge as variables that are components of total storm size but more explicitly linked to the flushing of hippo pools.” (Lines 442-445).